# New Definitions and Evaluations for Saliency Methods: Staying Intrinsic, Complete and Sound

**Arushi Gupta**[*,1]**, Nikunj Saunshi**[*,1]**, Dingli Yu**[*,1]**, Kaifeng Lyu**[1]**, Sanjeev Arora**[1]

[1]Princeton University
{arushig,nsaunshi,dingliy,klyu,arora}@cs.princeton.edu

[*]Denotes equal contribution

## Abstract

Saliency methods compute heat maps that highlight portions of an input that were most *important* for the label assigned to it by a deep net. Evaluations of saliency methods convert this heat map into a new *masked input* by retaining the $k$ highest-ranked pixels of the original input and replacing the rest with "uninformative" pixels, and checking if the net's output is mostly unchanged. This is usually seen as an *explanation* of the output, but the current paper highlights reasons why this inference of causality may be suspect. Inspired by logic concepts of *completeness & soundness*, it observes that the above type of evaluation focuses on completeness of the explanation, but ignores soundness. New evaluation metrics are introduced to capture both notions, while staying in an *intrinsic* framework—i.e., using the dataset and the net, but no separately trained nets, human evaluations, etc. A simple saliency method is described that matches or outperforms prior methods in the evaluations. Experiments also suggest new intrinsic justifications, based on soundness, for popular heuristic tricks such as TV regularization and upsampling.

## 1 Introduction

*Saliency methods* try to understand why the deep net gave a certain answer on a particular input, and are an important component of explainability, fairness, robustness, etc., in deep learning. This paper restricts attention to the (large) class of saliency methods that return an importance score for each coordinate of the input — often visualized as a heat map — which captures the coordinate's importance to the final decision.[1] Early saliency methods used an axiomatic system of "credit attribution" to individual input coordinates, using backpropagation-like methods [5, 36] and cooperative game theory, such as Shapley values [27, 45]. See Section 2 and Samek et al. [34] for discussion of strengths and limitations of such methods. More recent methods try to find heat maps that are largely concentrated on a smaller set of pixels/coordinates and are discussed further below.

There exists an ecosystem of *evaluation metrics* to evaluate saliency methods. Evaluations can be *extrinsic*, involving human evaluation [1], and comparison to certain ground truth explanations [47]. We restrict attention to *intrinsic* evaluations, which use computations involving the net itself and the heat/saliency map — but no human evaluations or evaluations that involve training new deep nets. Popular intrinsic evaluations include *saliency metric* [9] and insertion & deletion metrics [30].

A frequent idea in intrinsic evaluations (see Section 3 for references) is to create a new composite input — or sequence of such inputs — using the heat map and the original input, and to evaluate the original net on this composite input. For example, if $M$ is a binary vector with 1's in the $k$ coordinates with

---

[1]Heat maps suffice for recognition/classification tasks; other tasks may require more complex explanations.

36th Conference on Neural Information Processing Systems (NeurIPS 2022).

| Original Image | Masked Image for Cat | Masked Image for Ship | Original Image | Masked Image for Frog | Masked Image for Bird |
|:---:|:---:|:---:|:---:|:---:|:---:|
| 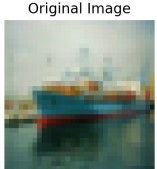 | 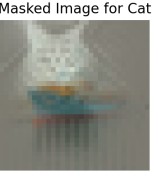 | 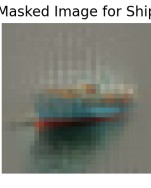 | 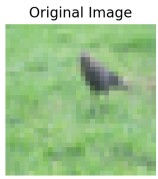 | 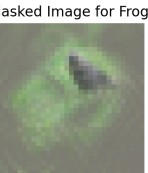 | 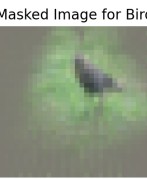 |

Figure 1: Masked CIFAR-10 images generated by our procedure with TV regularization. "Artifacts" exist for masks generated for incorrect labels; more examples can be found in Figure 11 in Appendix. The base model outputs the correct label on the original image (ship and bird resp.) with probability at least 0.99, and assigns probability at most $10^{-5}$ for the incorrect label (cat and frog resp.). With the generated masks, the AUC metric (defined in Section 3.1) for the correct label remains high (around 0.94 and 0.90), which corresponds to *completeness*, but AUC metric for the incorrect label rises tremendously (around 0.18 for cat mask, 0.71 for frog mask.) This suggests violation of *soundness*.

the highest values in the heat map, then $x \odot M$ (with $\odot$ denoting coordinate-wise multiplication) can be viewed as a masked input where only $k$ of the original coordinates of $x$ remain. It is customary to replace the zeroes in this masked input with "uninformative" values, which we refer to as *gray* pixels. This masked input is fed into the original net[2] to check if the net outputs the same label as on the full input — and, if so, one should conclude that the unmasked coordinates of $x$ were salient to the net's output. Not surprisingly, recent methods [11, 9, 31] for finding heatmaps use an objective that tries to directly optimize performance on such evaluations, greatly outperforming older axiom-based methods.

*Should we trust such mask-based explanations*? At first sight, this question may appear naive. In the above-described masked input $x \odot M$, some $k$ pixels were copied from the original image and the remaining pixels were made "gray." If the original net outputs essentially the same label as it did on the full image, does this not *prove* that the portion that came from the mask $M$ must have *caused* the net's output on the full image? The answer is "No", because the *positions* of the gray pixels can carry some signal.

Figure 1 shows that in standard datasets, it is possible to create masks out of most inputs such that the masked input can cause the net to assign high probability for a *different* label than in its original prediction. Prior works referred to this phenomenon as "mask artifacts" and suggested using total variation (TV) regularization to mitigate this issue. But Figure 1 shows TV regularization is not a full solution to masks that cause the net to (incorrectly) flip the answer.

**Soundness needed.** The masked input method is trying to *prove* that a certain portion of the image "caused" the net's output. Usually logical reasoning must display both *completeness* (i.e., all correct statements are provable) and *soundness* (incorrect statements cannot be proved). Checking that the net's output is essentially unchanged with the masked input is akin to verifying *completeness*.

But this by itself should not be convincing because, as mentioned, it is possible to use the method to find a different mask to justify another (i.e., wrong) label. *Soundness* would require verifying that the same saliency method cannot be used to produce masked inputs that make the net output a different label. Together, completeness and soundness ensure that the evaluation of the masked inputs produced using various labels approximately tracks the model's probability of assigning the label (see Section 3.2).

**Main Contributions of this paper:**

- Section 3.2 draws connections to logical proof systems and formalizes completeness and soundness in the context of saliency methods. These are intrinsic evaluations of saliency without appealing to human judgements (e.g., whether or not machine-generated masks contain artifacts). Our notion of completeness and soundness requires saliency methods to output a map for *every label* and not just the model prediction.

- Section 4 revisits methods that learn masks through optimization. Incorporating soundness, we propose a simple optimization method that maximizes the probability for any given label[3], rather than only the label deemed most likely by the net. A key design choice of this method is the pixel

---

[2]This raises a potential issue of distribution shift because the net never trained on masked inputs. In practice, especially for image data, trained nets continue to work fine on masked inputs.

[3]Computation cost proportional to the number of labels is only incurred while designing/evaluating the saliency method but not deployment time. Another alternative is to verify soundness condition for top $k$ labels.

replacement strategy during training: non-salient pixels are replaced with pixels of a random image, instead of using gray pixels, blurring or counterfactual models. [4]

- Experiments (in Section 6) estimate completeness and soundness scores for various methods. Our simple method does well on completeness and soundness as one would expect, but the slight surprise is that it performs comparably to prior masked-based methods on earlier saliency evaluations as well, suggesting that soundness can be achieved without paying much in completeness. Heuristic choices in existing mask-based methods such as TV regularization and upsampling, which were hitherto justified extrinsically (i.e., they make masks look natural to humans), receive an *intrinsic* and precise justification: they improve soundness. A theoretical result in Section 5 complements this experimental finding by showing that TV regularization helps saliency methods even in a simple linear classification setting, not just for vision data.

## 2 Prior approaches

We relegate a more thorough description of prior work to Appendix E, but mention some common saliency and saliency evaluation methods here. Saliency methods aim to explain a model's decision about an input. Saliency evaluation methods aim to evaluate the goodness of a saliency method.

**Saliency methods** include *backpropagation based approaches* such as Gradient $\odot$ Input [38], iGOS++ [20], Robust Perturbations [22], LRP [5], GradCAM [36], Smooth-Grad [39]. Another line of work is *masking methods* which include techniques based on averaging over randomly sampled masks [30], optimizing over meaningful mask perturbations [11], and real time image saliency using a masking network [9]. Pixels that have been removed from the image by the mask may be replaced by greying out, by Gaussian blurring as in [11], or with infillers such as CA-GAN [46] used in [8, 31], or DFNet [14]. The pixel replacement strategy we used is closely related to hot deck imputation [32], where features may be replaced either by using the mean feature value (analogous to replacing with grey) or sampling from the marginal feature distribution (analogous to replacing with image pixels sampled from other training images). Some prior work [7] has found that mean imputation does not significantly affect model output on the beer aroma review dataset. On Imagenette, by contrast, we found that replacement strategy can matter (See Figure 4). De Cao et al. [10] find masks using differentiable masking. *Boolean logic* is another approach for saliency methods [18, 17, 28, 29, 49].

**Arguments about saliency.** Discussions about the methods and the meaning of saliency appear, among others, in [37, 12, 13, 40]. Some reveal situations where Shapley axioms work against feature selection or where Shapley values may be calculated in conflicting ways [12, 40]. Others question efficacy of saliency methods that add noise [37], making explanations non class discriminative [13].

**Saliency evaluation methods.** Extrinsic evaluation metrics include the **WSOL** metric, and **Pointing Game** metric proposed by Zhang et al. [47] and **ROAR** [15]. Other more intrinsic methods include early saliency evaluation techniques like **MorF** and **LerF** [33], **Insertion and Deletion game** proposed by Petsiuk et al. [30], which involve either inserting pixels in order of most importance or deleting pixels in order of most importance. **BAM** [44] creates saliency maps by pasting object pixels from MSCOCO [25]. The **Saliency Metric** proposed by Dabkowski and Gal [9] thresholds saliency values above some $\alpha$ chosen on a holdout set, finds the smallest bounding box containing these pixels, upsamples and measures the ratio of bounding box area to model accuracy on the cropped image, $s(a, p) = \log(\max(a, 0.05)) - \log(p)$ where $a$ is the area of the bounding box and $p$ is the class probability of the upsampled image. Adebayo et al. [1] proposed "sanity checks" for saliency maps. They note that if a model's weights are randomized, it has not learned anything, and therefore the saliency map should not look coherent. They also randomize the labels of the dataset and argue that the saliency maps for a model trained on this scrambled data should be different than the saliency maps for the model trained on the original data. We study the effect of model layer randomization on our method in Appendix section D.4 and find that randomizing the model weights does cause our saliency maps to look incoherent. Tomsett et al. [43] also discover sanity checks for saliency metrics, finding that saliency evaluation methods can yield inconsistent results. They evaluate saliency maps on reliability, i.e. how consistent the saliency maps are. To measure a method's reliability, because access to ground truth saliency maps are not available, they use three proxies 1) inter-rater reliability, i.e. how whether a saliency evaluation metric is able to consistently rank some saliency methods

---

[4]Code for computing completeness, soundness, and our masking method available at `https://github.com/agup/soundness_saliency.git`

above others, 2) inter-method reliability, which indicates whether a saliency evaluation metric agrees across different saliency methods, and 3)internal consistency reliability, which measure whether different saliency methods are measuring the same underlying concept.

**Discussions.** Brunke et al. [6] show that perturbation methods are sensitive to baseline and Petsiuk et al. [30] point out that human centric explanations (based on bounding boxes) may not reveal why the model made a certain decision. Our notion of intrinsic saliency method outputs the mask but introduces soundness to validate it.

## 3 Masking explanations and completeness/soundness

The goal of a saliency method is to produce explanations for the outputs of a given deep net $f$. We think of a saliency method (Definition 3.1) producing explanations — a heat map in this case — as providing a *proof* of the statement $f$ *outputs the label $a$ for input $x$*. Inspired by logical proof systems, we propose that intrinsic evaluations for saliency methods should test for both completeness and soundness, and the rest of the section provides definitions for these notions.

### 3.1 Saliency methods and AUC metric

For a net $f$, let $f(x, a)$ denote the (post-softmax) output probability of the net on input $x$ for label $a$. We are interested in saliency methods that output a heatmap with a scalar per coordinate of the input.

**Definition 3.1** (Saliency Method). A saliency method $\mathcal{M}$ is an algorithm that takes 1) a deep net $f$, 2) an input $x$, and 3) a label $a$ and produces a heatmap $\boldsymbol{m} = \mathcal{M}(f, x, a) \in \mathbb{R}^{\dim(x)}$.

Note that in the above definition, $\mathcal{M}$ can access any part of $f$ including intermediate layers and gradients. Examples of methods that can be interpreted as returning heatmaps are mask-based methods [11, 9, 8, 2, 31] that output heatmaps in $[0, 1]^{\dim(x)}$, backpropagation-like methods [5, 36], Shapely values [27, 45], and Gradient $\odot$ Input.

Given such a saliency method, an important question is how to evaluate the quality of the "explanations" or heat maps it produces. A common evaluation idea is to use the behavior of the net $f$ on a "masked" input — only the top few pixels based on the heat map in question are retained, while the others are hidden appropriately. Inspired by this idea we define evaluation metrics for saliency methods. We abstract the step of retaining the top few pixels into an *input modification process*, defined below.

**Definition 3.2** (Input Modification Process $\Gamma$). Let $\Gamma$ be a potentially randomized procedure that takes an input $x$ and a heatmap $\boldsymbol{m}$ and generates a modified input $\tilde{x}$. i.e. $\tilde{x} \sim \Gamma(x, \boldsymbol{m})$.

This input modification process defines a *base metric* that is the starting point of the completeness and soundness metrics that will be defined later.

**Definition 3.3** (Base Metric). Let $x$ be an input (image), $a$ a label, $\boldsymbol{m}$ a heatmap and $\Gamma$ an input modification process. Then, a base metric is a function $g(x, a, \boldsymbol{m}) = \mathbb{E}_{\tilde{x} \sim \Gamma(x, \boldsymbol{m})}[f(\tilde{x}, a)]$, where $f$ is the neural net of interest[5].

The base metric measures the expected output of $f$ acting on *modified* inputs $\tilde{x}$ and label $a$ and different choices for $\Gamma$ lead to different base metrics. Some examples of $\Gamma$ that stay in the *intrinsic* framework are: graying out the pixels outside some subset $S$ of $\boldsymbol{m}$, replacing them by pixels from a Gaussian blurring of $x$ [11]. Input modification through $\Gamma$ also shows up in saliency methods, including our proposed method in Section 4; we discuss prior choices of $\Gamma$ in that section. For our evaluation metrics, however, we choose $\Gamma$ that grays out remaining pixels.

The popular Area-Under-the-Curve (AUC) evaluation metrics [30] of saliency methods can, in fact, be reinterpreted in terms of our base metric. We state the *insertion game* and its AUC below, since it forms the basis of our subsequent completeness and soundness metrics, and then define the AUC metric as an instantiation of Definition 3.3.

**Definition 3.4** (AUC of insertion game). For $s = 1$ to $\dim(x)$ take the top $s$ pixels as per $\boldsymbol{m}$, and plot the probability $f(x, a)$ given by model to label $a$ on the input $x$, where the top $s$ pixels of $x$ are retained and remaining pixels are assigned a default gray value. Return the area under the curve.

---

[5]In general we can replace $f$ with any target function $h$ that depends on $f$

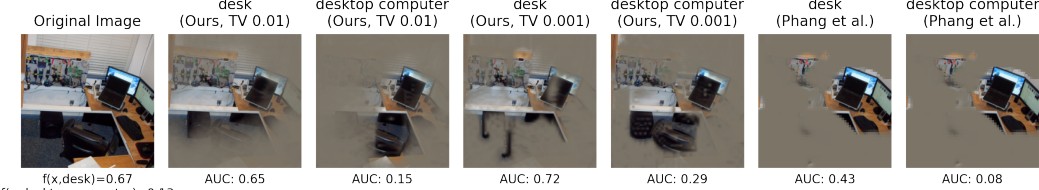

Figure 2: ImageNet image and masked versions for labels desk and desktop computer. Heatmaps generated by our method with $\lambda_{TV} = 0.01$, $\lambda_{TV} = 0.001$ and Phang et al. [31]'s method, respectively.

**Definition 3.5** (AUC metric). We denote by $g_{\text{AUC}}(x, a, \boldsymbol{m})$ the base metric when a modified input $\tilde{x} \sim \Gamma(x, \boldsymbol{m})$ is obtained as follows:

- Sample $s \sim \{1, \ldots, \dim(x)\}$ and pick $S$ to be top $s$ pixels as per $\boldsymbol{m}$.
- Set $\tilde{x}[S] = x[S]$ and $\tilde{x}[\bar{S}] = x_{\text{gray}}[\bar{S}]$ where $x_{\text{gray}}$ is a "default" input with gray values.

We note that $g_{\text{AUC}}$ is indeed equivalent to the AUC insertion game from Definition 3.4. While many choices of base metrics can be used, we use this AUC metric for subsequent definitions since it confers two benefits: 1) it allows for fair comparison between methods since all methods are evaluated at the same sparsity level; 2) it considers the effect of multiple sparsity levels instead of picking an arbitrary one. We are now ready to present the formalization of completeness and soundness.

### 3.2 Completeness and soundness

Just as for proof systems, we would like a saliency method to be logically complete and sound, i.e., it should be able to justify the label output by the net, but no other label[6]. The statements to be "proved" are of the form *"does $f$ think $x$ has label $a$?"*. We use the model output probability $f(x, a)$ as fractional truth value of the "statement" $(x, a)$. For a saliency map $\boldsymbol{m}$ that a saliency method $\mathcal{M}$ generates, we interpret the AUC metric $g_{\text{AUC}}(x, a, \boldsymbol{m})$ as the evaluation of truth value of the "proof" $\boldsymbol{m}$, as adjudged by the base metric $g_{\text{AUC}}$. Thus in this context, completeness for $\mathcal{M}$ would entail that whenever $f(x, a)$ is large (i.e., the statement is sufficiently true), $g_{\text{AUC}}(x, a, \boldsymbol{m})$ should be large (a sufficient good proof can, and is, generated). Soundness, on the other hand, would entail that $g_{\text{AUC}}(x, a, \boldsymbol{m})$ should be small whenever $f(x, a)$ is small. Thus we would like $g_{\text{AUC}}(x, a, \boldsymbol{m})$ to track the value of $f(x, a)$ through a lower bound and upper bound as follows:

**Definition 3.6** (Completeness and Soundness). Consider a saliency method $\mathcal{M}$. For $\alpha, \beta \leq 1$, the method $\mathcal{M}$ is $\alpha$-*complete on* $f, x, a$ if the heatmap $\boldsymbol{m} = \mathcal{M}(f, x, a)$ it produces satisfies $\boxed{g_{\text{AUC}}(x, a, \boldsymbol{m}) \geq \alpha \, f(x, a)}$, while $\mathcal{M}$ is $\beta$-*sound on* $f, x, a$ if $\boxed{g_{\text{AUC}}(x, a, \boldsymbol{m}) \leq \frac{1}{\beta} f(x, a)}$. Conversely the completeness and soundness scores for a heap map $\boldsymbol{m}$ on $(x, a)$ are defined as:

$$\alpha(x, a) = \min\left\{\frac{\max\{g_{\text{AUC}}(x, a, \boldsymbol{m}), \epsilon_1\}}{f(x, a)}, 1\right\}, \quad \beta(x, a) = \min\left\{\frac{\max\{f(x, a), \epsilon_2\}}{g_{\text{AUC}}(x, a, \boldsymbol{m})}, 1\right\}. \quad (1)$$

**Definition 3.7** (Worst case completeness and soundness). For a saliency method $\mathcal{M}$, these metrics are defined by taking the worst case of the completeness/soundness over labels $a$ for a given $x$ that is sampled from the underlying input distribution. More precisely, $\alpha(\mathcal{M}) = \mathbb{E}_x\left[\min_a \alpha(x, a)\right], \beta(\mathcal{M}) = \mathbb{E}_x\left[\min_\beta \beta(x, a)\right]$.

Our metrics implicitly require a saliency method to output a meaningful map for every label (or at least all labels assigned non-negligible probabilities by the net) and not just for the top model prediction. Notice, $\alpha$-completeness means that if the model outputs a high probability for label $a$, then the probability for label $a$ after seeing only the coordinates in the salient sets is also high, while $\beta$-soundness means that this probability is not too high. Checking both conditions verifies if the AUC score $g_{\text{AUC}}(x, a, \boldsymbol{m})$ of a map $\boldsymbol{m}$ is in the interval $\left[\alpha f(x, a), \frac{1}{\beta} f(x, a)\right]$. We want $\alpha, \beta$ as close to 1 as possible. This notion is different from the usual AUC insertion game, which just requires $g_{\text{AUC}}(x, \hat{y}, \boldsymbol{m})$ to be as large as possible for the model prediction $\hat{y} = \arg\max_a f(x, a)$. Moreover,

---

[6]We focus on single label multi-class classification; extensions to multi-label are left for future work.

our worst-case completeness and soundness metrics require the $g_{\text{AUC}}(x, a, \boldsymbol{m})$ to roughly *match* the model prediction $f(x, a)$, *for every pair* $(x, a)$,[7] where the thresholds $\epsilon_1, \epsilon_2$ in Equation (1) help in ignoring the case where the model outputs very small probabilities — for instance we can ignore any label for completeness if $f(x, a) < \epsilon_1$.

### 3.3  Illustration of completeness and soundness on an ImageNet example

See Figure 2. For this image $x$, the model assigns reasonably high probabilities to labels $a_1 = $ "desk" and $a_2 = $ "desktop computer", with $f(x, a_1) \approx 0.67$ and $f(x, a_2) \approx 0.13$. We compare the maps computed by three methods — $\mathcal{M}_1$: our mask learning procedure from Section 4 with TV regularization 0.01, $\mathcal{M}_2$: our method with TV of 0.001, and $\mathcal{M}_3$: the mask learned by Phang et al. [31]. The corresponding maps generated are denoted by $\boldsymbol{m}_i, i \in \{1, 2, 3\}$. For this discussion we only consider the effect of labels $a_1$ and $a_2$ on completeness and soundness, and we assume the thresholds satisfy $\epsilon_1 = \epsilon_2 = 0$.

**Completeness.**   For the label $a_1$, the AUC scores $g_{\text{AUC}}(x, a_1, \boldsymbol{m}_i)$ for the three methods are roughly 0.65, 0.70 and 0.43 respectively. Thus all three maps can "certify" the label $a_1$ well enough, with $\boldsymbol{m}_2$ doing the best job. Consequently the completeness scores (Equation (1)) for $\boldsymbol{m}_1$ on the pair $(x, a_1)$ is $\alpha_1(x, a_1) = \min\{\frac{0.65}{0.67}, 1\} \approx 0.97$; similarly $\alpha_2(x, a_1) = 1$ and $\alpha_3(x, a_1) = \frac{0.43}{0.67} \approx 0.64$. Similarly for label $a_2$ that is assigned a probability of 0.13, the completeness scores $\alpha_i(x, a_2)$ are 1, 1, and 0.58. Thus the worst case completeness from Definition 3.7, i.e. $\min_{j \in \{1, 2\}} \alpha(x, a_j)$, for the three methods are 0.97, 1 and 0.58. Based on completeness $\mathcal{M}_2$ seems like the best method; however soundness will paint a different picture.

**Soundness.**   Using the same AUC scores we compute the soundness scores, defined in Equation (1) as $\beta_i(x, a) = \min\{\frac{f(x, a)}{g_{\text{AUC}}(x, a, \boldsymbol{m}_i)}, 1\}$. This gives us soundness scores on $a_1$ of 1, 0.95 and 1. The more interesting case is label $a_2$, for which the model does not assign very high probability (0.13). Here the mask from $\mathcal{M}_2$ still gets a very high AUC score of 0.29, leading to a soundness score $\beta_2(x, a_2) \approx \frac{0.13}{0.29} \approx 0.44$. $\mathcal{M}_1$ on the other hand is not as overconfident in its explanation as $\mathcal{M}_2$, and gets $\beta_1(x, a_2) = \frac{0.13}{0.15} \approx 0.89$. The worst case soundness for three methods ends up being 0.89, 0.44 and 1.

This examples highlights how completeness and soundness together can give a more nuanced comparisons of saliency maps, including understanding the benefit of TV regularization. Experiments in Section 6 do a more thorough comparison of many saliency methods on these and prior metrics.

## 4   Procedures for finding masking explanations

We propose a very simple method to find saliency maps with good empirical completeness and soundness scores. Detailed intuitions and implementations appear in Appendix A. Our method is similar to prior work on mask-based saliency maps and is based on the idea of SSR (Smallest Sufficient Region)[8]. In particular for an input $x$ and label $a$, given a network $f$, the main goal is to learn a map (or mask) $M \in \{0, 1\}^{hw}$ such that $\mathbb{E}_{\tilde{x} \sim \Gamma(x, M)}[-\log(f(\tilde{x}, a))]$ is minimized, i.e. the probability that the network assigns to a modified (or composite) input $\tilde{x}$ is high. The key difference from prior masked-based methods is our choice of $\Gamma$, which retains the pixels of $x$ corresponding to $M$, but replaces the rest of the pixels with values from a *randomly sampled* image $\bar{x}$ from the training set[9] $\mathcal{X}$, i.e.

$$\tilde{x} \sim \Gamma(x, a) \equiv \bar{x} \sim \mathcal{X}, \tilde{x} = M \odot x + (1 - M) \odot \bar{x} \tag{2}$$

Modifications procedures ($\Gamma$) that have been considered in prior work include: graying or blurring out the remaining pixels [11], and using a conditional image generative model to fill in the remaining pixels [8, 2]. Replacing with random image pixels forces the mask-finding algorithm to solve a more difficult task of having the network predict a high confidence for the label, despite the presence of another image. This amounts to grafting salient pixels of $x$ on top of a random image, reminiscent of BAM

---

[7]Works like Gradient $\odot$ Input and LRP use *completeness* in a somewhat different sense. It requires the sum of the coordinate scores in the heatmap to *exactly* equal the logit of the label. Our completeness + soundness together try to approximately match the $g_{\text{AUC}}$ score to the output probability on this label.

[8]We do not incorporate any SDR (Smallest Destroying Region) component in our method.

[9]We think other random image distributions should work too.

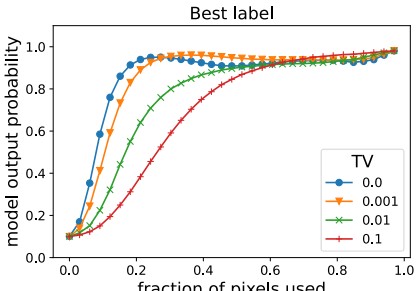 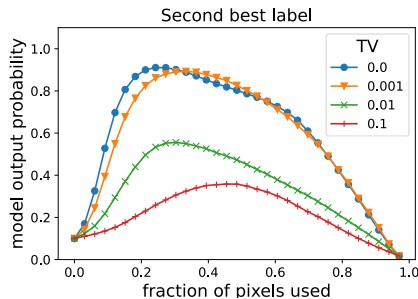

Figure 3: Plot of model output probability as more pixels from the original image are retained using learned masks. The remaining pixels are replaced with gray. Different curves correspond to different values of TV regularization ($\lambda_{TV}$). Larger Area-Under-the-Curve (AUC) for the left figure (best label) suggests good completeness, while lower AUC for the right figure suggests good soundness. Plots suggest that adding TV significantly helps with soundness, while only slightly hurting completeness.

evaluations [44] for saliency methods. We also note a methodological difference from Dabkowski and Gal [9], Phang et al. [31] in that we do not train a separate neural network to output the mask.

As is standard, we relax the domain of masks $M$ from binary $\{0,1\}^{hw}$ to continuous $[0,1]^{hw}$ parametrized by a sigmoid. We define a composite input, where the part of $x$ on $M$ is superimposed onto a distractor $\bar{x} \sim \mathcal{X}$ as $\tilde{x} = M \odot x + (1-M) \odot \bar{x}$. We then have the following simple *vanilla* objective, where the $\ell_1$ norm penalty on $M$ helps to reduce the size of masks.

$$L(M;(x,a)) = \mathbb{E}_{\bar{x} \sim \mathcal{X}}\left[-\log(f(\tilde{x},a))\right] + \lambda_1 \|M\|_1, \text{ where } \tilde{x} = M \odot x + (1-M) \odot \bar{x}. \quad (3)$$

**Promoting soundness.** Tricks used in prior mask-finding approaches were tested for their effect on soundness. These include 1) Total-Variation (TV) penalty [11] and 2) upsampling of the mask from a lower resolution one [30] by learning a low-resolution mask at scale $s$, $M \in [0,1]^{hw/s^2}$. For example, on ImageNet, $s=1$ corresponds to learning a $224 \times 224$ mask, while $s=4$ corresponds to learning a $56 \times 56$ mask and upsampling by a factor of 4. Incorporating 1) and 2) leads to the following *modified* objective

$$L(M;(x,a)) = \mathbb{E}_{\bar{x} \sim \mathcal{X}}\left[-\log(f(\tilde{x},a))\right] + \lambda_{TV} TV(M^{\times s}) + \lambda_1 \|M^{\times s}\|_1$$
$$\text{where } \tilde{x} = M^{\times s} \odot x + (1-M^{\times s}) \odot \bar{x}, \quad (4)$$

and $M^{\times s} \in \mathbb{R}^{hw}$ is obtained by upsampling $M$ by a factor of $s \in \{1,4\}$ via bilinear interpolation.

While the motivation cited for these two commonly employed "tricks" is to avoid artifacts [11], in our intrinsic framework "artifacts" does not have an interpretation. What looks like an artifact to a human observer may be relevant to the net's decision-making. Indeed, our experiments show that while TV penalty or upsampling does produce better looking masks, they lead to a drop in the *completeness metric*. But they significantly improve *soundness*. This provides an intrinsic justification for the use of such tricks, instead of extrinsic justifications related to human interpretability.

## 5 On benefit of TV regularization in saliency

Past justifications for TV regularization in saliency methods focused on its ability to make heatmaps in images that look more natural to humans. But that is not an intrinsic justification. In the intrinsic framework proposed in this paper, experiments in Section 6 show TV regularization helps improve soundness of the saliency method. The current section reinforces this intrinsic benefit of TV by sketching a simple example (details in Appendix F) showing how TV helps even in a simple linear model $\boldsymbol{x} \mapsto \boldsymbol{w}^\top \boldsymbol{x}$ on non-image input $\boldsymbol{x} \in \mathbb{R}^d$ with binary label $y \in \{\pm 1\}$.

Suppose that the classifier has margin at least $\gamma > 0$, that is, $y\boldsymbol{w}^\top \boldsymbol{x} = \sum_{i=1}^d yw_i x_i \geq \gamma$. We focus on saliency methods that can return a binary heatmap $\boldsymbol{m} \in \{0,1\}^d$ to certify the label, where the coordinates marked as 1 constitute a set $S$ of saliency coordinates. Let $\Gamma(x, \boldsymbol{m})$ be the input modification process that assigns 0 to all input coordinates outside the corresponding saliency set $S$.

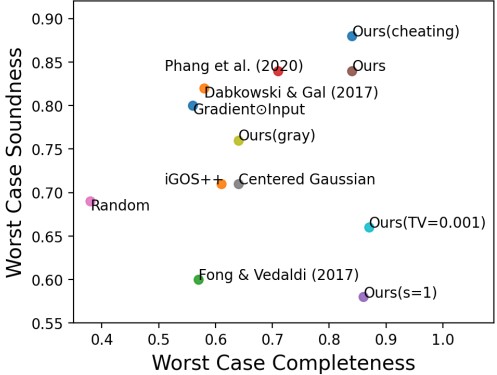 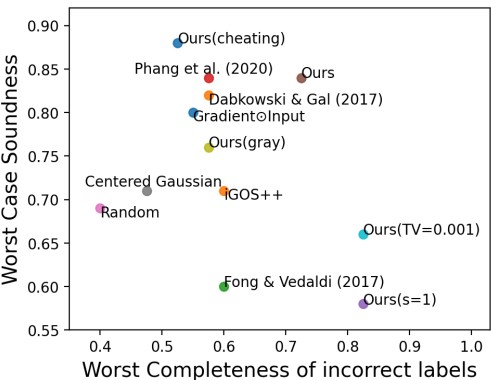

Figure 4: Worst case completeness v.s. soundness plot on Imagenette for different methods. (Upper right corner is best.) Each point represent either a prior method [31, 9, 11, 38] or our method with different settings. By default, our methods use upsampling $s = 4$, TV penalty $\lambda_{TV} = 0.001$ and use other images for composite image (see Equation (2)). "Ours" uses the default setting; "Ours($s = 1$)" uses upsampling $s = 1$; "Ours($TV = 0.001$)" uses TV penalty factor $\lambda_{TV} = 0.001$; "Ours(gray)" fills gray pixels in modifications procedures $\Gamma$; and "Ours(cheating)" uses masks generated for the correct label as masks for all labels. The $x$-axis of the right figure is the average worst completeness of all incorrect labels over samples where the second best label has model probability at least 0.01. See interpretations of the plot in Section 6.2.

Consider an input $x$ such that $\boldsymbol{w}^\top \boldsymbol{x} > 0$ and $y = 1$. Firstly note that any set $S$ of coordinates that contribute positively to $\boldsymbol{w}^\top \boldsymbol{x}$, i.e., the base metric $g(x, y, \boldsymbol{m}) := \mathbb{1}_{[\sum_{i \in S} w_i x_i > 0]}$ is 1, will be able to certify the label $y = 1$ and thus would satisfy *completeness*. However this might not be convincing to a user, since it is also possible to come up with a set $S'$ that certifies the label $y = -1$ by picking only negative coordinates of $x$. This would not satisfy *soundness* as the original model only predicts $y = 1$.

This issue of soundness can be solved by adding a regularization or constraint on the set of allowable maps, in a way that ensures that it is easy to find a mask $\boldsymbol{m}$ for the correct label $y$, but not for $-y$. Ideally, the constraints should be task-specific, but for linear classification with no special structure, even a TV constraint after random permutation suffices. We consider the constraint that salient sets $S$ should be an interval; this corresponds to $\text{TV}(\boldsymbol{m}) \leq 2$. The following result shows that if we further constrain the length of intervals to be $\Theta(\frac{1}{\gamma^2} \log d)$, then with high probability, we can find an interval certifying the model prediction $y$ but we are unable to do the same for $-y$ (Details in Appendix F.)

**Theorem 5.1.** *For $(\boldsymbol{x}, y) \in \mathbb{R}^d \times \{\pm 1\}$ with $\|\boldsymbol{x}\|_2 = 1$, after random shuffling of the coordinates, the following holds for any $L_1 = \Omega(\frac{1}{\gamma^2} \log \frac{1}{\delta})$, $L_2 = \Omega(\frac{1}{\gamma^2} \log \frac{d}{\delta})$:*

1. *(Completeness) With probability $1 - \delta$, there is an interval $\boldsymbol{m}$ of length $L_1$ s.t. $g(x, y, \boldsymbol{m}) = 1$;*

2. *(Soundness) With probability $1 - \delta$, $g(x, -y, \boldsymbol{m}) = 0$ holds for all intervals $\boldsymbol{m}$ of length $L_2$.*

Although this example is simple, the conceptual message is clear: saliency methods may not guarantee soundness in itself, but adding regularity constraints such as TV can improve soundness.

## 6 Experiments

In this section, we aim to 1) show TV penalty and upsampling have intrinsic justification that they significantly improve soundness at a small cost in completeness; 2) evaluate and compare prior saliency methods, and our method with different parameter settings, on our completeness and soundness metrics. Experiments are performed either on CIFAR-10 or Imagenette [16], a 10-class subset of ImageNet [10]. We compute maps and evaluate metrics on 1000 randomly drawn images from the original test set. Additional experiments on various models and datasets (including ImageNet and

---

[10]Completeness and soundness evaluation on all labels can get computationally expensive on ImageNet as it contains 1000 classes. This cost is only relevant when designing the method; not at deployment time.

Table 1: Saliency methods evaluate on prior metrics, worst case completeness and soundness on Imagenette. Insertion calculated with grey infilling. The best two of each column are marked bold. By default, our methods use upsampling $s = 4$, TV penalty factor $\lambda_{TV} = 0.01$ and use randomly sampled image in the modification procedures $\Gamma$. Difference from default setting is noted in parentheses. $\downarrow$ indicates lower is better. *Strong performance of is likely due to centrality bias (cf. Appendix C).

| | Deletion $\downarrow$ | Insertion (gray) $\uparrow$ | Saliency Metric $\downarrow$ | Worst Case Completeness $\uparrow$ | Worst Case Soundness $\uparrow$ |
|---|---|---|---|---|---|
| Gradient $\odot$ Input[38] | 0.42 | 0.56 | $-0.35$ | 0.56 | 0.80 |
| iGOS++[20] | **0.37** | 0.68 | $-0.36$ | 0.61 | 0.71 |
| Dabkowski and Gal [9] | 0.48 | 0.66 | $-0.85$ | 0.58 | 0.82 |
| Fong and Vedaldi [11] | 0.58 | 0.59 | $-0.40$ | 0.57 | 0.60 |
| Phang et al. [31] | **0.41** | 0.75 | $-0.27$ | 0.71 | **0.84** |
| Ours | 0.50 | 0.74 | $-0.90$ | 0.84 | **0.84** |
| Ours ($s = 1$) | 0.49 | **0.77** | **$-1.18$** | **0.86** | 0.58 |
| Ours ($\lambda_{TV} = 0.001$) | 0.45 | **0.80** | **$-1.01$** | **0.87** | 0.65 |
| Ours (gray infilling) | 0.46 | 0.65 | $-0.35$ | 0.64 | 0.76 |
| Random | 0.45 | 0.45 | $-0.35$ | 0.38 | 0.69 |
| Centered Gaussian | 0.66 | 0.74* | $-0.97$* | 0.64 | 0.71 |

CIFAR-100) appear in Appendix D, as do details of training procedure (beyond those in Section 4). For all our completeness and soundness evaluations, we use thresholds of $\epsilon_1 = 0.01$ and $\epsilon_2 = 0.001$ (see Equation (1)).

## 6.1 Effect of TV penalty and upsampling

In this section, we study the effect of "tricks" like TV penalty and upsampling on the learned maps and evaluation metrics. Figure 3 shows how soundness and completeness of our masks changes with TV regularization. It plots the model output probability for CIFAR-10 as more pixels from the original image are retained using the mask. Our high level finding is that adding TV penalty significantly aids soundness, while only slightly hurting completeness. Exact evaluation of soundness and completeness for different $\lambda_{TV}$ are listed in Table 2 in Appendix C. Relevant points of Figure 4 (which also contain other saliency methods) show effect of upsampling and TV penalty on completeness and soundness. We compare our method from Section 4 with different settings on Imagenette. Our method by default use $s = 4$ and $\lambda_{TV} = 0.01$. By comparing "Ours" against "Ours($s = 1$)" and "Ours($TV = 0.001$)", we observe that upsampling and TV penalty leads to slightly worse completeness but significantly better soundness. The effect of upsampling on previous saliency evaluation metric may be seen in the relevant entries of Table 1, where it leads to slightly worse performance on the insertion, deletion, and saliency metrics.

## 6.2 Comparison to existing metrics and methods

We compared our procedure to other methods including Gradient $\odot$ Input [38], iGOS++[20], Dabkowski and Gal [9] Fong and Vedaldi [11] and Phang et al. [31] on Imagenette. We also include simple baselines like "Random" map (each entry is a random Gaussian), and "Centered Gaussian" map (2-dimensional isotropic Gaussian distribution placed at the image center). Besides completeness and soundness metrics, we evaluate methods on the Deletion Game and Insertion Game metrics (`https://github.com/eclique/RISE`), and Saliency metric (SM) [9]. Detailed description of the different metrics can be found in the Appendix E.

For our method, we learn a map by optimizing the objective function in Equation (4) with $\lambda_{TV} \in \{0.01, 0.001\}$ and $s \in \{1, 4\}$; default settings are $\lambda_{TV} = 0.01, s = 4$. We also evaluate our method when other pixels are replaced with gray rather than a random image, denoted by "Ours (gray)". For uniformity, we use the identical ResNet50 pretrained on ImageNet as the base classifier for every saliency method. We normalize the maps so that all values lie in $[0, 1]$ before use.

**Completeness and soundness.** We compare different methods on the metrics from Definition 3.7, through the visualization in Figure 4 (left) and in Table 1. We find that our method achieves better completeness and good soundness compared to other methods and baselines. We posit that completeness is better due to our improved composite input strategy from Equation (2), where non-salient pixels are filled with another image pixels as opposed to other strategies. As an ablation, we see that "Ours (gray)" which uses gray replacement does not perform as well.

On the other hand, our soundness is good only for higher TV regularization $\lambda_{TV} = 0.01$ and higher upsampling $s = 4$, as discussed in Section 6.1. Soundness is very bad (but completeness slightly better) if these "tricks" are not used. We note that while Phang et al. [31] does as well as our method on soundness, it does so via "cheating" in an interesting way: they compute mask for the top label and return the same mask for all labels. Since the method does not even try to generate masks for labels with low probability, it achieves good soundness for free. To delve deeper, we employ the same "cheating" with our method — return the same mask for all labels — and observe a large bump in soundness, without any drop in completeness. While "cheating" is always possible (even if unintentionally) for any metric, we hope that our results inspire saliency method designers to compute maps for all labels. In this case we find, in Figure 4 (right), that looking at worst case completeness of incorrect labels (all except model prediction) gives a measure of the "effort" a method employs in generating masks for all labels with probability at least 0.01. As evident methods that "cheat", like Phang et al. [31] and "Ours (cheating)", achieve a bad "best effort" score.

**Other metrics.** Table 1 suggests that our method can achieve comparable performances on most existing intrinsic metrics. This suggests that good completeness and soundness scores are achievable without compromising on other measures. Deletion metric is one where our method falls short, likely due to our method not employing an SDR objective as in Dabkowski and Gal [9]. Although we note that "Random" baseline performs quite well on the Deletion metric, which suggests that deletion metric does not provide much signal on these datasets. Further investigation on the good performance of "Centered Gaussian" baseline can be found in Appendix C.

Our settings that works best on other metrics are those without upsampling ($s = 1$) or lower TV ($\lambda_{TV} = 0.001$). Soundness is the only metric that strongly justifies the upsampling and TV "tricks", thus verifying its utility. Our method with $s = 4, \lambda_{TV} = 0.01$ demonstrates that good performance can be obtained on all metrics simultaneously, with only little price to pay.

**Limitations of our work.** Completeness and soundness do not involve or address human interpretability of explanations. This can be a strength when considering intrinsic evaluations, but is a weakness when considering extrinsic evaluations. Additionally, though completeness and soundness should be taken into account while designing a saliency method, they are not meant as a replacement for other evaluation methods, which can provide additional information on saliency map quality. For example, the axioms and sanity checks for saliency discussed in Appendix E.1 should also be applied. Finally, at deployment time, there is a risk that these evaluation methods may be used as a "proof" to justify a saliency method is safe to be used in high-risk applications. Completeness and soundness should be taken as a sanity check rather than a proof a saliency method is safe for deployment.

## 7 Conclusions

Saliency explanations of ML models have proved nebulous and generated much discussion. By avoiding extrinsic considerations (e.g., human interpretability) and sticking to intrinsic notions such as completeness and soundness, this paper has tried to give a rigorous notion of saliency that is intrinsic to the deep net, while avoiding the noisiness in earlier intrinsic ideas that motivated methods like Gradient⊙Input and Shapley Values. Other new contributions include clarifying in the intrinsic view the role of TV regularization and other tricks (it hurts completeness slightly but greatly improves soundness); a simple saliency method (Section 4) for producing mask-based explanations that was designed solely with intrinsic considerations and yet has performance competitive with good existing methods, and sometimes better (Table 1). Our experiments suggest soundness provides a new dimension to evaluate saliency methods on, and that good performance on it can be achieved without compromising on other metrics. Note that soundness needs to be considered only while designing and evaluating the method, not at deployment.

Evaluating soundness requires looking at maps for all labels instead of just the top label, and thus seems relevant for object *localization* in images, and for classification in the wild — where multiple objects appear in an image. This benefit is hinted in our evaluation on a prior small dataset (see Appendix B) and deserves further exploration. It may help with certain distribution shifts as well. While this work studies soundness in the context of mask-based saliency methods and evaluations, the concept of soundness could be applicable for the idea of saliency and interpretability in more generality.

**Acknowledgements.** We thank Ruth Fong for feedback on an earlier draft of this paper. We are also grateful to the valuable comments from various anonymous reviewers that helped improve the paper. This work is supported by funding from NSF, ONR, Simons Foundation, DARPA and SRC.

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
