# A Intuitions and Implementations of Procedures to Find Masking Explanations

As introduced in Section 3.1, for evaluation we may interest in random binary masks due to its connection to AUC, but in our method for finding masking explanations we only focus on deterministic masks. Given a network $f$, image $x \in \mathbb{R}^{c \times hw}$ and class $a$, we wish to find a binary mask $M \in \{0, 1\}^{hw}$ such that when the part of $x$ on $M$ is superimposed onto a "distractor" $\bar{x} \sim \mathcal{X}$ (randomly sampled from the train set) as $\tilde{x} = M \odot x + (1 - M) \odot \bar{x}$, the output probability of the model $f(\tilde{x}, a)$ is high for the class $a$.

As in Section 3.2 we compute the average probability assigned to class $a$ over the sampling of the distractor $\bar{x}$, i.e. we are interested in making $\mathbb{E}_{\bar{x} \sim \mathcal{X}}[f(\tilde{x}, a)]$ high. To avoid the hard problem of optimizing over the hypercube $\{0, 1\}^{hw}$, a typical strategy (also employed in prior work) is to relax the domain of $M$ to be $[0, 1]^{hw}$. Since we do not wish to learn masks of very large size, a $\ell_1$ norm penalty on $M$ (corresponding to size of the mask), leading to the following natural objective function[11]

$$L(M) = \mathbb{E}_{\bar{x} \sim \mathcal{X}} \left[ -\log(f(M \odot x + (1 - M) \odot \bar{x}, a)) \right] + \lambda_1 \|M\|_1 \tag{5}$$

However most masking-based methods employ additional "tricks" in order to avoid "artifacts" in the produced saliency maps, like Total-Variation (TV) penalty [11] and upsampling of the mask from a lower resolution one [30]. We also employ the same strategy by learning a low-resolution mask at scale $s$, $M \in \mathbb{R}^{hw/s^2}$, to minimize the following

$$L(M) = \mathbb{E}_{\bar{x} \sim \mathcal{X}} \left[ -\log(f(M^{\times s} \odot x + (1 - M^{\times s}) \odot \bar{x}, a)) \right] + \lambda_{TV} TV(M^{\times s}) + \lambda_1 \|M^{\times s}\|_1 \tag{6}$$

where $M^{\times s} \in \mathbb{R}^{hw}$ is obtained by upsampling $M$ by a factor of $s \in \{1, 4\}$ via bilinear interpolation.

While the motivation cited for these "trick" is to avoid artifacts, it is not clear whether artifacts are a bad thing, since they might be relevant to the net's decision-making. Indeed, we show that while TV penalty or upsampling does produce better looking masks, they lead to a drop in the *completeness metric*. However we show that adding such tricks leads to significant improvement in the *soundness metric,* thus providing a novel justification for the use of such tricks, beyond just the heuristic argument of getting rid of artifacts. In Section 5 we also provide theoretical justification for why TV penalty can help with soundness, even for the simple case of linear predictors on non-image data.

We optimize the objective in Equation (4) by parametrizing $M$ as a sigmoid of real valued weights $W \in \mathbb{R}^{hw/s^2}$, i.e. $M = \sigma(W)$, and run Adam [23] optimizer for 2000 steps with learning rate 0.05 and by sampling 10 distractor images at every step, for different values of $\lambda_{TV}$ and upsampling factor $s$.

# B Practical Benefits of Completeness of all labels for Images of Multiple Objects

Images may have multiple plausible labels. Figure 5 and Figure 6 show images where the classifier net gave high probability to a single label even though multiple objects were present. Our saliency method can produce different and meaningful masks for all labels that are valid. Extending the notion of soundness for multi-label settings is an open question.

In Figure 5, images that previously used in Gu et al. [13] can have both elephants and zebras present, but it may not be always clear from the model output if there is such a case, since the model can be much more confident on one label, e.g., elephant, than one would expect it to be. For this reason, finding masking explanations validating other labels, e.g., zebra, could provide more information on how the model makes the prediction.

We also use the relabeling provided by Beyer et al. [4] to select ImageNet validation images with two true labels. Our results may be seen in Figure 6.

---

[11]A standard way to maximize probability is to minimize the negative log probability



Figure 5: Images containing both elephant(s) and zebra(s), and the corresponding masked ones generated by our method and the best-performing CA model in Phang et al. [31]. The masks by Phang et al. [31] are identical for different labels, and contains both elephant and zebra. In contrast, our method outputs decent masks for elephant and zebra accordingly. For more examples please see Figure 6 in Appendix C.

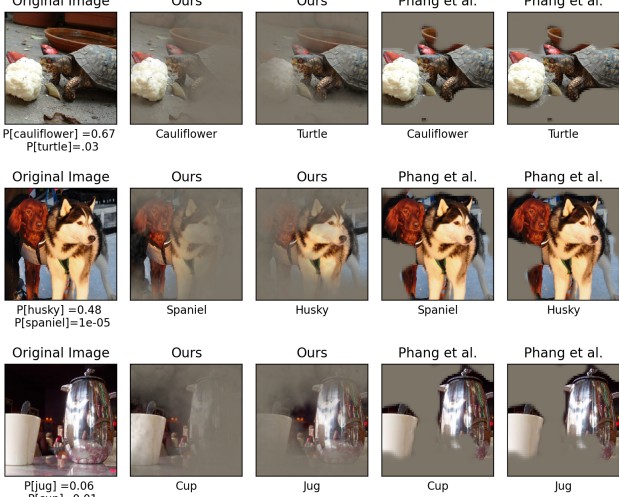

Figure 6: Our masks and masks by Phang et al. [31] for ImageNet images with two ground truth labels. The best-performing CA model in Phang et al. [31] was used. First (leftmost) column depicts original image with original model probabilities for each ground truth class below the image. Next two columns depict our masks with target label below the image. Final two columns depict the masks for Phang et al. [31]. The masks by Phang et al. [31] are identical for different labels, and contains both classes. In contrast, our method outputs descent masks for each class accordingly.

Table 2: Worst case completeness and worst case soundness ($\epsilon_1 = 0.01, \epsilon_2 = 0.1$) for different $\lambda_{TV}$ in CIFAR-10. The best one in each row is marked bold.

| $\lambda_{TV}$ | 0 | 0.001 | 0.01 | 0.1 |
|---|---|---|---|---|
| Completeness | **0.99** | 0.99 | 0.89 | 0.80 |
| Worst soundness | 0.10 | 0.10 | 0.11 | **0.19** |

Table 3: Deletion, insertion, saliency metric, worst case completeness and soundness on Imagenette-Corner. Our method with upsampling factor $s = 4$, TV penalty $\lambda_{TV} = 0.01$ achieves the best completeness and soundness. ($\uparrow$ indicates higher is better.) The numbers for Centered Gaussian drop compared to Imagenette.

| | Deletion $\downarrow$ | Insertion (gray) $\uparrow$ | Saliency Metric $\downarrow$ | Worst Case Completeness $\uparrow$ | Worst Case Soundness $\uparrow$ |
|---|---|---|---|---|---|
| Gradient $\odot$ Input | **0.13** | 0.34 | $-$**0.25** | 0.50 | 0.78 |
| Dabkowski and Gal [9] | 0.21 | 0.33 | $-$**0.25** | 0.52 | 0.78 |
| Fong and Vedaldi [11] | **0.15** | 0.53 | $-$**0.25** | 0.53 | 0.48 |
| Phang et al. [31] | 0.33 | **0.69** | $-$**0.25** | **0.68** | **0.82** |
| Ours | 0.36 | **0.64** | $-$**0.25** | **0.76** | **0.84** |
| Random | 0.35 | 0.35 | $-$0.24 | 0.36 | 0.64 |
| Centered Gaussian | 0.62 | 0.62 | 0.458 | 0.50 | 0.54 |

## C  Additional Results on CIFAR-10 and Imagenette

In this section, we show additional results for CIFAR-10 and Imagenette experiments.

1. In Table 2, we show the worst case completeness and worst case soundness on CIFAR-10 of our method with different TV penalty $\lambda_{TV}$. To showcase of generality of our definition of base metric, here we use a modified version of AUC metric where we set $s \sim \{0.2 \dim(x), \ldots, 0.6 \dim(x)\}$ in Definition 3.5.

2. We showcase the masks of ten randomly drawn images from Imagenette for different methods in Figure 7 and Figure 8. The masks in Figure 7 is generated for model predictions, while the masks in Figure 8 is generated for incorrect labels. Note there are some wired horizontal lines and shapes for some masks. These are caused by default tie breaking of pixels of masking value 1. We also tried tie breaking according to Centered Gaussian, which does not improve the performance of those methods.

3. We create a new dataset Imagenette-Corner based on Imagenette, where each image is a random corner of the original image[12]. We show the results on Imagenette-Corner in Table 3. The results show that the good performance of Centered Gaussian on Imagenette is likely due to the bias of datasets (that objects are centered with high probability). Saliency metric on Imagenette-Corner are the same for most of the methods, likely because of the coarse selection of hyperparameters as in Dabkowski and Gal [9].

## D  Experimental Details and Additional Experiments

In this section we expand upon the experiments in Section 6 and complement them with more experiments on the ImageNet, CIFAR-10 and CIFAR-100 datasets. For each of the datasets we test the following:

- **Visualization:** For various values of TV regularization (and upsampling for ImageNet), we visualize the mask and also what part of the image a sparse version of the mask highlights. We do so for masks learned for the correct label and also for the second most probable label as predicted by the model. The common trend is that while TV regularization (and upsampling) make the masks more human interpretable, it also makes it harder to find a good mask for the incorrect label, thus improving soundness.

---

[12]On ImageNet or Imagenette, the common way to process image is first resizing it to $256 \times 256$ pixels, and then crop the center $224 \times 224$ pixels. In Imagenette-Corner, instead of center cropping, we take one of the four $180 \times 180$-pixel corners and resize it to $224 \times 224$ pixels.

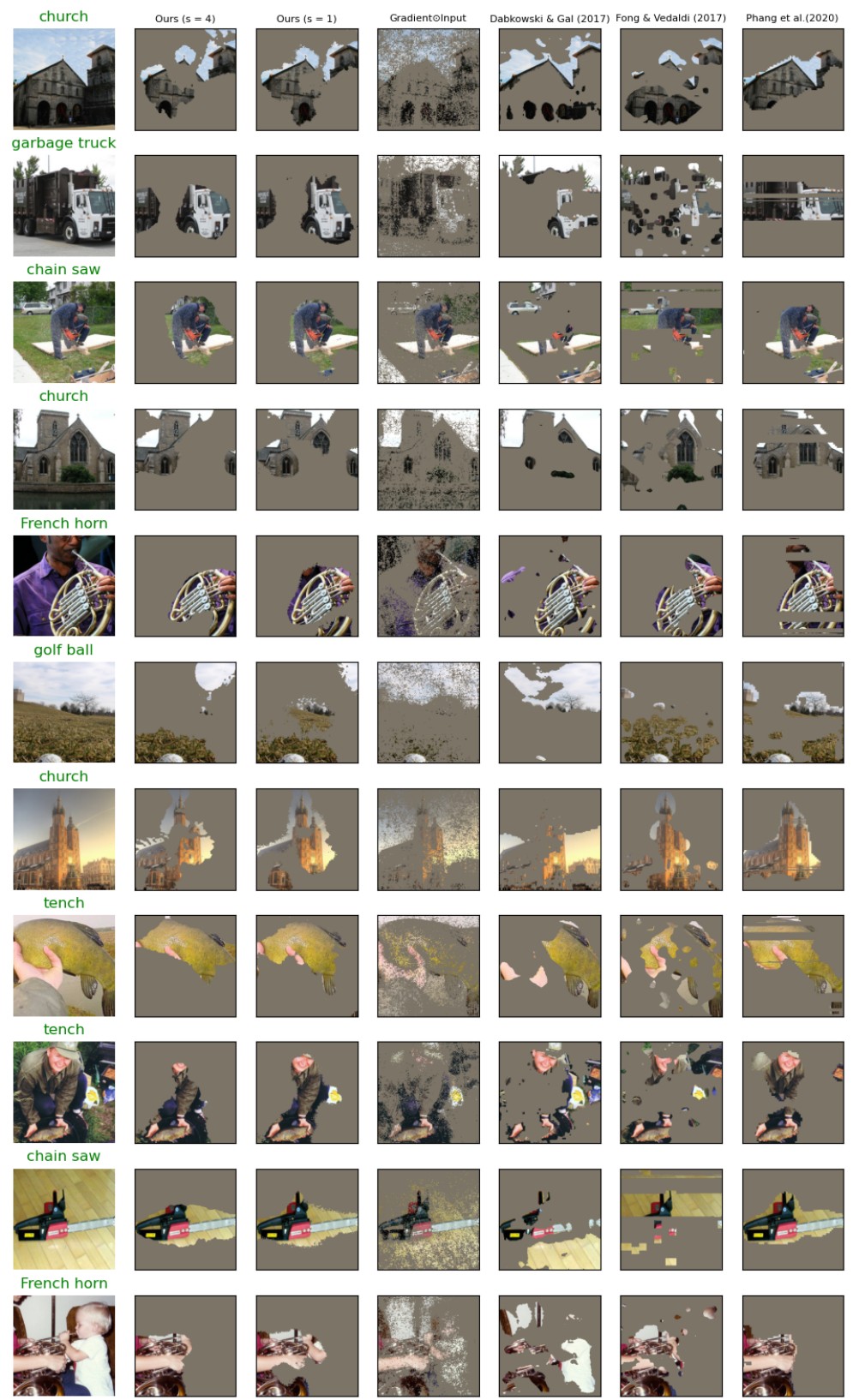

Figure 7: Masks of 10 Imagenette examples generated by different methods. Above each original image is the corresponding correct label. 30% of the pixels are retained, and the rest pixels are filled with grey.

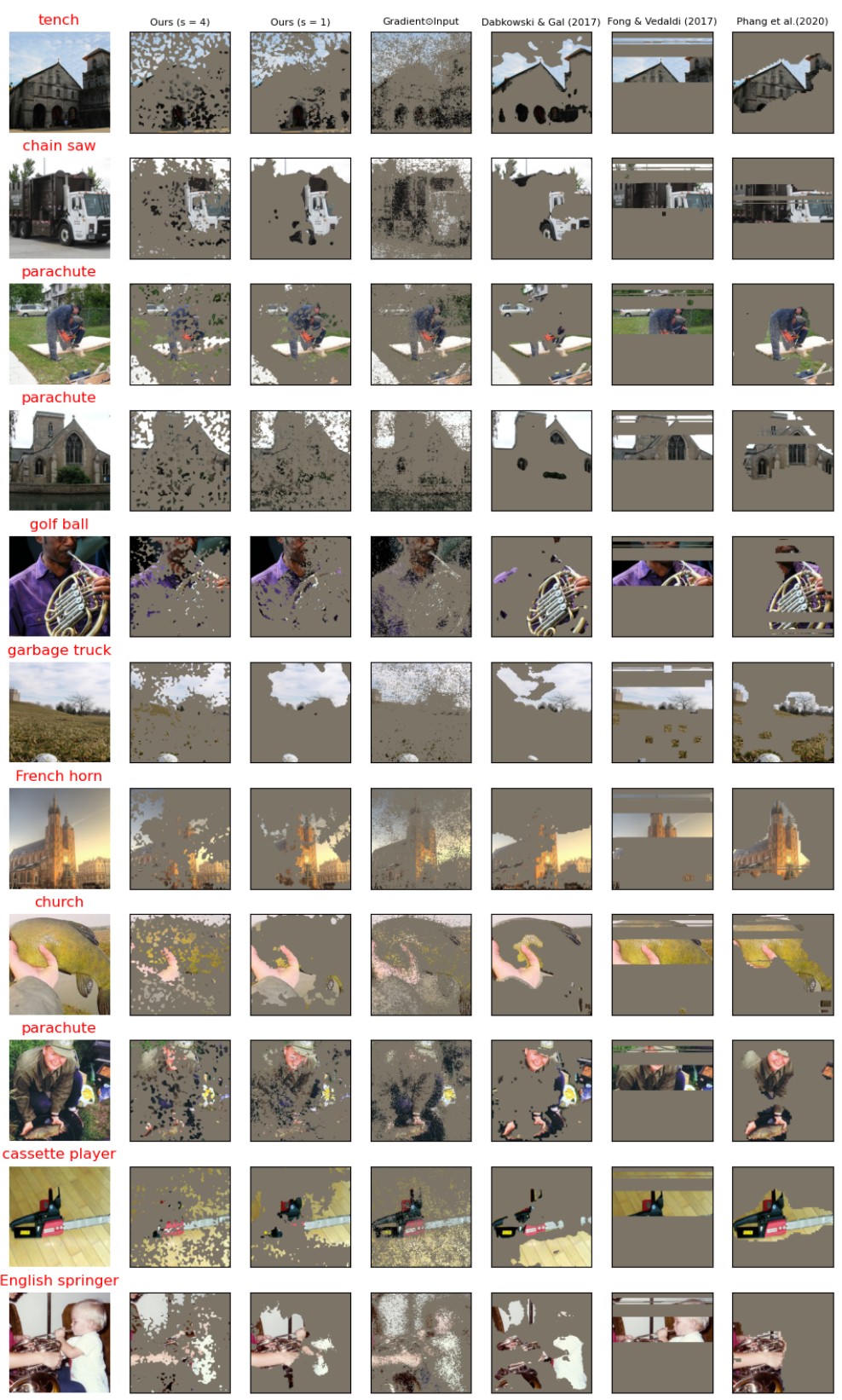

Figure 8: Masks of 10 Imagenette examples generated by different methods for incorrect labels. Above each original image is the target incorrect label. 30% of the pixels are retained, and the rest pixels are filled with grey.

Table 4: Completeness and soundness for a ResNet-164 trained model on CIFAR-10, as defined in Equation (1). Each column represents mask learned using our procedure, with different TV regularization strengths (0.0, 0.001, 0.01, or 0.1). "Gray" indicates pixels were grayed during AUC calculation and "Random image" indicates they were replaced with other images.

| Gray | TV = 0.0 | TV= 0.001 | TV= 0.01 | TV= 0.1 |
|---|---|---|---|---|
| Completeness ($\alpha$) | 0.92 | 0.91 | 0.83 | 0.77 |
| Soundness ($\beta$) | 0.17 | 0.18 | 0.39 | 0.57 |
| Random image | TV = 0.0 | TV= 0.001 | TV= 0.01 | TV= 0.1 |
| Completeness ($\alpha$) | 0.86 | 0.85 | 0.77 | 0.70 |
| Soundness ($\beta$) | 0.20 | 0.21 | 0.44 | 0.62 |

- **AUC curve:** We plot the output model probability for a masked input as more pixels from the original image are selected. The 4 plots denote replacing remaining pixels with gray pixels or pixels from a random image, and masks to fit the correct or incorrect labels, i.e. most probable and second most probable labels. Again, we find the TV regularization and upsampling help with soundness; i.e. inability to find mask for the second most probable label. For mask $M$, if $\bar{M}(p)$ denotes the discrete mask with top $p$ fraction of the pixels from $M$ picked. We plot $\mathbb{E}_x \left[ \mathbb{E}_{x' \sim \Gamma}[f(\bar{M}(p) \odot x + (1 - \bar{M}(p)) \odot x', a)] \right]$ v/s $p$, where $\Gamma$ is either a random image or a gray image, and $a$ is either the correct label for $x$ or the second best label. We note that replacing with gray and random image lead to similar looking plots, with the probability estimate of random image being more pessimistic. This justifies the motivation for our procedure that learns a mask to solve a "harder task" of random image replacement.

- **Completeness/soundness:** We evaluate the completeness and soundness scores, as defined in Equation (1) and Definition 3.7. In particular for any input $x$, we only evaluate the scores for the top model prediction $a$ and the second most probable label $a'$ and report the worst case completeness and soundness for these 2 labels. For all experiments in this section, we use $\epsilon_1 = 0$ and $\epsilon_2 = 0.1$ (from Equation (1)).

- **Intrinsic metrics:** We evaluate our masks on other intrinsic metrics from prior work, and compare to baseline saliency methods. Our baselines include Gradient $\odot$ Input [38], Smooth-Grad [39], Real Time Saliency [9] (for ResNet-50 on ImageNet), and Random indicating a random Gaussian mask as a control. We use Captum [24] for Gradient $\odot$ Input and Smooth-Grad implementations and the original author code[13] for Real Time Saliency. When calculating the Saliency Metric (SM) [9] we tune the threshold $\delta$ on a holdout set of size 100 with $\delta$ between 0 and 5 in increments of 0.2 as in prior work.

  For the saliency method of Fong and Vedaldi [11] that we only used on the Imagenette, we adapt the most popular implementation on GitHub[14]. The implementation contains minor deviations from the original paper as described on its main page. For Phang et al. [31], we used their best CA model pretrained and provided in original author code[15].

## D.1 CIFAR-10 Experiments

We also run our method from Section 4 on the CIFAR-10 dataset using a pretrained ResNet-164 architecture[16]. For all experiments we learn a mask $M \in \mathbb{R}^{32 \times 32}$, thus using a scaling factor of $s = 1$ (no upsampling). We train masks for 1600 images that were correctly classified by the pretrained ResNet-164 using regularization parameter $\lambda_{TV} \in \{0, 0.001, 0.01, 0.1\}$. We use a (fixed) L1 regularization value of .001 for all masks.

We visualize the masks learned for the correct label in Figure 10a and in Figure 10b we visualize the same for the second best label predicted by the ResNet-164 model. We also visualize the masks for all labels for some randomly picked images in Figure 11 to demonstrate the commonness of artifact,

---

[13] https://github.com/PiotrDabkowski/pytorch-saliency
[14] https://github.com/jacobgil/pytorch-explain-black-box
[15] https://github.com/zphang/saliency_investigation
[16] https://github.com/bearpaw/pytorch-classification. The ResNet-110 model in this repository is actually a ResNet-164 model.

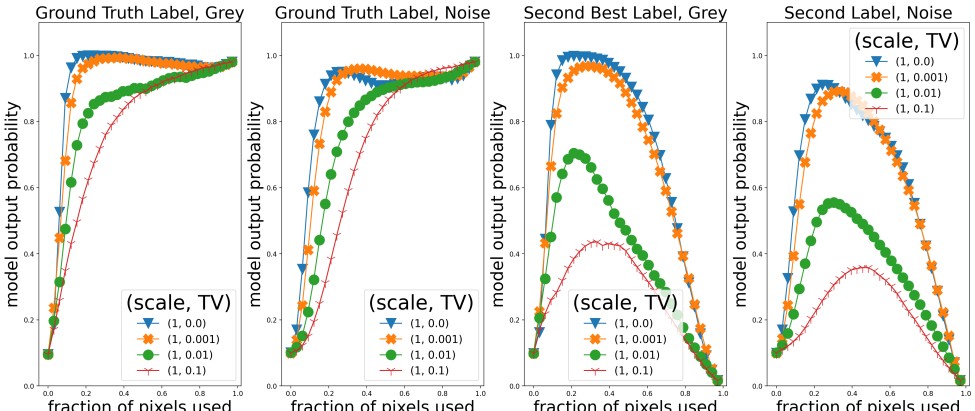

Figure 9: [CIFAR-10] AUC curves with as the fraction of pixels retained from the original images based on the mask varies from 0 to 1.0 on the X-axis. The probabilities assigned by the model (averaged over 1600 images) on the Y-axis. **Left:** Mask learned for ground truth label, probabilities for ground truth label while replacing remaining pixels with gray. **Center Left:** Mask learned for ground truth label, probabilities for ground truth label while replacing remaining pixels with other image pixels. **Center Right:** Mask learned for second best label, probabilities for second best label while replacing remaining pixels with gray. **Right:** Mask learned for second best label, probabilities for second best label while replacing remaining pixels with other image pixels. We see that increasing TV regularization results in only a mild drop in completeness, but significantly improves soundness.

Table 5: Performance of our method on CIFAR-10 and some baselines on various intrinsic saliency metrics proposed in prior work. Downarrow (uparrow) means lower (higher) is better. We find that while both our masks (learned with and without TV) have very good performance on the insertion metric. The deletion and saliency metrics are uninformative in this case, since all methods are as good (or worse) compared to a random mask.

|  | Gradient $\odot$ Input | Our method ($\lambda_{TV} = 0.01$) | Our Method ($\lambda_{TV} = 0$) | Smooth-Grad saliency | Random |
|---|---|---|---|---|---|
| Deletion $\downarrow$ | 0.32 | 0.37 | 0.59 | 0.31 | 0.26 |
| Insertion (blur) $\uparrow$ | 0.60 | 0.88 | 0.94 | 0.66 | 0.36 |
| Insertion (gray) $\uparrow$ | 0.51 | 0.83 | 0.92 | 0.55 | 0.26 |
| Saliency Metric $\downarrow$ | 0.22 | 0.22 | 0.22 | 0.23 | 0.22 |

especially for the incorrect labels. The AUC curves in Figure 9 suggest a similar trend to that of Imagenette, adding TV regularization results in only a mild drop in completeness, but significantly improves soundness. Evaluation of our masks, compared to some gradient baselines, on intrinsic metrics can be found in Table 5. We report the completeness and soundness results for CIFAR-10 in Table 4 for TV values in $(0.0, 0.001, 0.01, 0.1)$ calculated using a ResNet-164 model.

### D.2 CIFAR-100 Experiments

We run the same experiment for CIFAR-100 using the corresponding ResNet164 model. We visualize the masks learned for the correct label in Figure 13a and in Figure 13b we visualize the same for the second best label predicted by the ResNet-164 model. The AUC curves in Figure 12 suggest a similar trend to that of Imagenette, adding TV regularization results in only a mild drop in completeness, but significantly improves soundness. Evaluation of our masks, compared to some gradient baselines, on intrinsic metrics can be found in Table 7. We place a downarrow after the name of the metric to indicate a lower value is considered better and an uparrow when a higher value is considered better. We evaluate on a randomly selected subset of 1600 data points where the model had correct top 1 accuracy. We report the completeness and soundness results for CIFAR-100 in Table 6 for TV values in $(0.0, 0.001, 0.01, 0.1)$ calculated using a ResNet-164 model.

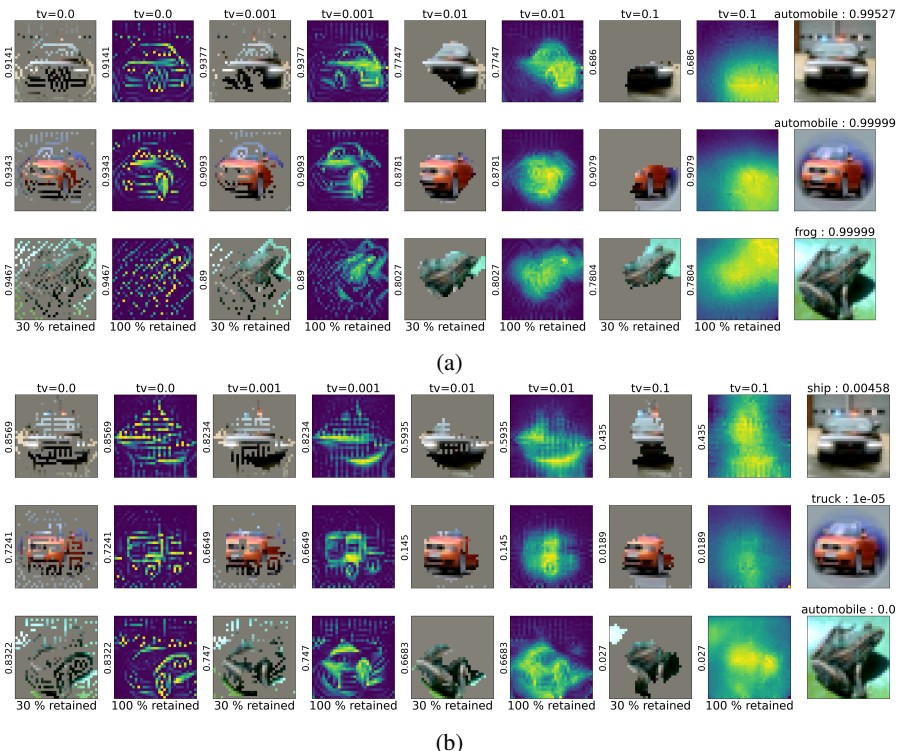

(a)

(b)

Figure 10: Details in Appendix D.1 **Panel 10a** Masks learned for the correct label of CIFAR-10 images using the procedure outlined in Section 4 with ResNet-164. Columns (1,3,5,7) depict masked images at 30 (retained) % mask sparseness. Columns (2,4,6,8) depict the original mask. TV values shown above. Original image shown in rightmost column. Model probability of correct label for masked images on y axis. **Panel 10b** Masks leared for the second most probable label of CIFAR-10 images using the procedure outlined in Section 4 on ResNet-164. Columns (1,3,5,7) depict masked images at 30 % mask sparseness. Columns (2,4,6,8) depict the original mask.TV values shown above. Original image shown in rightmost column. Model probability of second best label for masked images on y axis.

Table 6: Completeness and soundness for a ResNet-164 trained model on CIFAR-100, as defined in Equation (1). Each column represents mask learned using our procedure, with different TV regularization strengths (0.0, 0.001, 0.01, or 0.1). "Gray" indicates pixels were grayed during AUC calculation and "Random image" indicates they were replaced with other images.

| Gray | TV = 0.0 | TV= 0.001 | TV= 0.01 | TV= 0.1 |
|---|---|---|---|---|
| Completeness ($\alpha$) | 0.80 | 0.74 | 0.64 | 0.55 |
| Soundness ($\beta$) | 0.28 | 0.34 | 0.58 | 0.75 |
| Random image | TV = 0.0 | TV= 0.001 | TV= 0.01 | TV= 0.1 |
| Completeness ($\alpha$) | 0.67 | 0.66 | 0.61 | 0.54 |
| Soundness ($\beta$) | 0.35 | 0.40 | 0.61 | 0.78 |

### D.3 Experiments on ImageNet

In Figure 14a we depict the the masks for TV values in $\{0.0, 0.01\}$ for a ResNet-18 model on ImageNet for the ground truth label and in Figure 14b we depict the same for the second best label. We also experiment with the effect of upsampling (US) the mask, whereby we learn a mask of size (56,56) and upsample to size (224,224). We use a fixed L1 regularization value of 2e-5. We depict our results on ImageNet and ResNet-18 in Table 9.

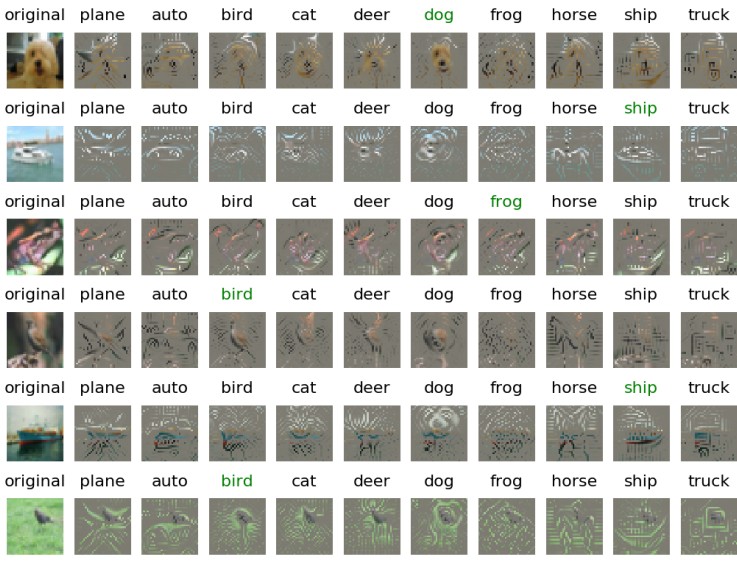

(a) Our method with no TV regularization

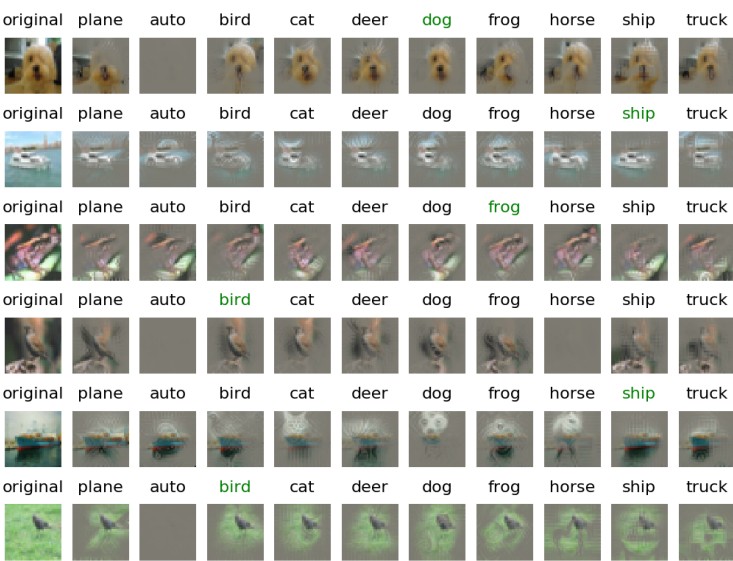

(b) Our method with TV regularization $\lambda_{TV} = 0.01$

Figure 11: A demonstration of artifacts created by masking on CIFAR-10. Pixels (partially) masked out are filled with gray based on the fractions they are masked out. Masks generated without or only with low level regularization can easily produce artifacts. It is more common and/or severe for the incorrect label than correct label.

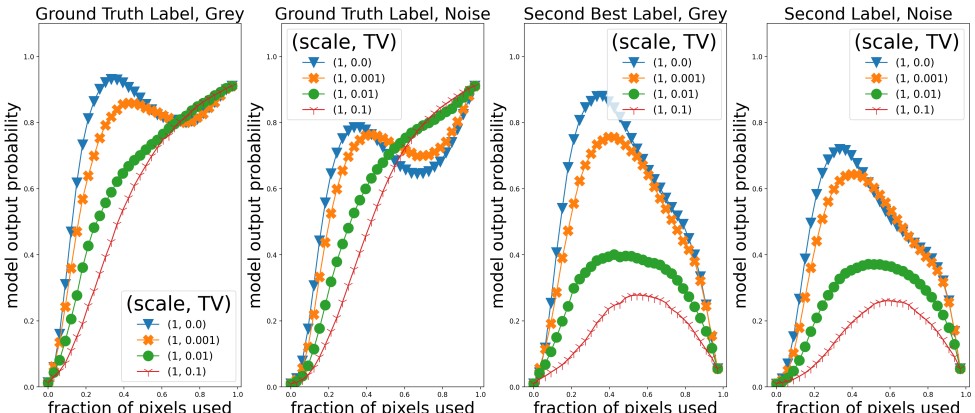

Figure 12: [CIFAR-100] AUC curves with as the fraction of pixels retained from the original images based on the mask varies from 0 to 1.0 on the X-axis. The probabilities assigned by the model (averaged over 1600 images) on the Y-axis. **Left:** Mask learned for ground truth label, probabilities for ground truth label while replacing remaining pixels with gray. **Center Left:** Mask learned for ground truth label, probabilities for ground truth label while replacing remaining pixels with other image pixels. **Center Right:** Mask learned for second best label, probabilities for second best label while replacing remaining pixels with gray. **Right:** Mask learned for second best label, probabilities for second best label while replacing remaining pixels with other image pixels. We see that increasing TV regularization results in only a mild drop in completeness, but significantly improves soundness.

Table 7: Performance of our method on CIFAR-100 and some baselines on various intrinsic saliency metrics proposed in prior work. Downarrow (uparrow) means lower (higher) is better. We find that while both our masks (learned with and without TV) have very good performance on the insertion metric. The deletion and saliency metrics are uninformative in this case, since all methods are as good (or worse) compared to a random mask.

| | Gradient $\odot$ Input | Our method $(\lambda_{TV} = 0.01)$ | Our Method $(\lambda_{TV} = 0)$ | Smooth-Grad saliency | Random |
|---|---|---|---|---|---|
| Deletion $\downarrow$ | 0.10 | 0.17 | 0.10 | 0.29 | 0.11 |
| Insertion (blur) $\uparrow$ | 0.36 | 0.71 | 0.82 | 0.39 | 0.20 |
| Insertion (gray) $\uparrow$ | 0.27 | 0.62 | 0.76 | 0.29 | 0.11 |
| Saliency Metric $\downarrow$ | 0.77 | 0.77 | 0.77 | 0.79 | 0.77 |

For the deletion metric, we note that most methods have comparable or worse performance than the random mask, which suggests that the metric does not give us much signal about the goodness of the saliency maps. On the insertion metric, we find that mask learned by not adding the TV penalty significantly beats other methods. The mask learned using TV penalty, on the other hand, has impressive performance on both the insertion AUC and saliency metric (SM).

**Completeness and Soundness on ImageNet and ResNet-18.** We report our results in Table 8 for TV values in $(0, 0.01)$ for both graying (Gray) and replacing with other image pixels (Random image). Additionally, we investigate the effect of upsampling (US) where we derive a $(56, 56)$ and upsample by a factor of 4 to a $(224, 224)$ mask.

**Effect of ensembling.** In order to investigate the effect of ensembling we plot maps in Figure 16 as we vary the number of maps that are ensembled over as $K \in \{1, 2, 4\}$, where we learn multiple masks such that each of them individually validate the label, but are as disjoint as possible. We do not upsample (using a scale of 1.0) and we use a fixed L1 regularization of 2e-5 and a fixed TV regularization of 0.0.

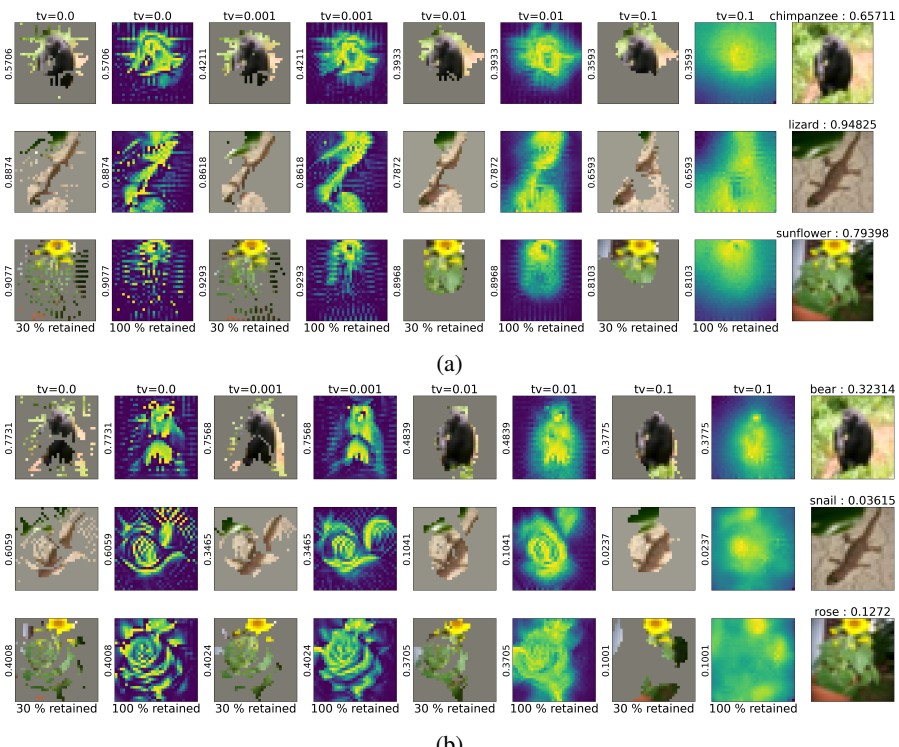

(a)

(b)

Figure 13: Details in Appendix D.2 **Panel 13a** Masks learned for the correct label of CIFAR-100 images using the procedure outlined in Section 4 on ResNet-164. Columns (1,3,5,7) depict masked images at 30 (retained) % mask sparseness. Columns (2,4,6,8) depict the original mask. TV values shown above. Original image shown in rightmost column. Model probability of correct label for masked images on y axis. **Panel 13b** Masks learned for the second most probable label of CIFAR-100 images using the procedure outlined in Section 4 on ResNet-164. Columns (1,3,5,7) depict masked images at 30 % mask sparseness. Columns (2,4,6,8) depict the original mask.TV values shown above. Original image shown in rightmost column. Model probability of second best label for masked images on y axis.

Table 8: Completeness and soundness for a ResNet-18 model on ImageNet as defined in Equation (1). Each column contains a represents mask learned using our procedure, with our without upscaling and different TV regularization strengths. "Gray" indicates pixels were grayed during AUC calculation and "Random image" indicates they were replaced with other images. No US indicates the full (224,224) mask was derived and US indicates a (56, 56) mask was derived then upsampled by a factor of 4. TV indicates a TV regularization value of 0.0 or 0.01.

| Gray | TV = 0.0 | TV = 0.01 | US TV = 0.0 | US TV = 0.01 |
|---|---|---|---|---|
| Completeness ($\alpha$) | 0.97 | 0.76 | 0.87 | 0.71 |
| Soundness ($\beta$) | 0.19 | 0.70 | 0.38 | 0.75 |
| Random image | TV = 0.0 | TV = 0.01 | US TV = 0.0 | US TV = 0.01 |
| Completeness ($\alpha$) | 0.89 | 0.61 | 0.74 | 0.59 |
| Soundness ($\beta$) | 0.25 | 0.83 | 0.52 | 0.86 |

### D.4 Effect on Sanity Checks

Inspired by [1] we randomize the last layer of a ResNet-18 network and visually inspect the resulting saliency maps in Figure 15. We find that the maps appear less coherent than those of a pre-trained model. We use a fixed L1 regularization of 2e-5 and depict maps with and without upsampling (US) at TV values of $(0, 0.01)$.

Table 9: Performance of our method on ImageNet and ResNet-18 model and some baselines on various intrinsic saliency metrics proposed in prior work. We find that while both our masks (learned with and without TV) have very good performance on the insertion metric, the mask learned with TV has much better performance on the saliency metric. The deletion metric is uninformative in most cases, since most methods are as good (or worse) compared to a random mask.

|  | Gradient $\odot$ Input | Our method $(\lambda_{TV} = 0.01)$ | Our Method $(\lambda_{TV} = 0)$ | Smooth-Grad saliency | Random |
|---|---|---|---|---|---|
| Deletion $\downarrow$ | 0.10 | 0.13 | 0.21 | 0.08 | 0.13 |
| Insertion (blur) $\uparrow$ | 0.44 | 0.79 | 0.85 | 0.51 | 0.31 |
| Insertion (gray) $\uparrow$ | 0.30 | 0.67 | 0.92 | 0.35 | 0.13 |
| Saliency Metric $\downarrow$ | 0.31 | 0.15 | 0.32 | 0.32 | 0.32 |

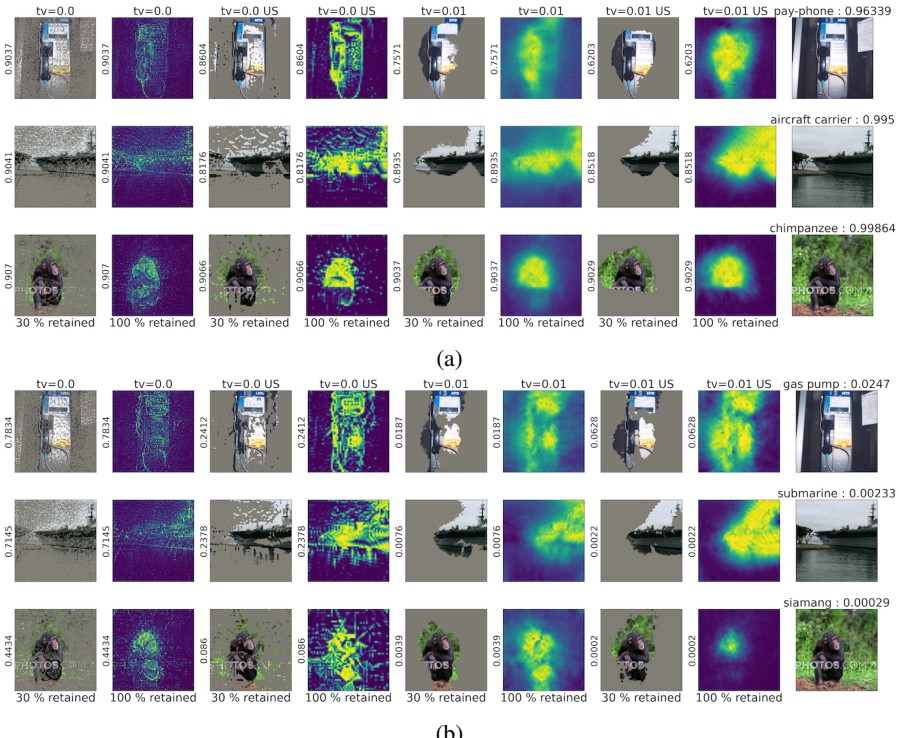

(a)

(b)

Figure 14: Details in Appendix D.3. US stands for upsampled mask, where we derive a (56,56) mask and interpolate to (224,224). **Panel 14a** Masks learned for the correct label of ImageNet images using the procedure outlined in Section 4 on ResNet-50. Columns (1,3,5,7) depict masked images at 30 (retained) % mask sparseness. Columns (2,4,6,8) depict the original mask. TV values shown above. Original image shown in rightmost column. Model probability of correct label for masked images on y axis. **Panel 14b** Masks learned for the second most probable label of ImageNet images using the procedure outlined in Section 4 on ResNet-50. Columns (1,3,5,7) depict masked images at 30 % mask sparseness. Columns (2,4,6,8) depict the original mask. TV values shown above. Original image shown in rightmost column. Model probability of second best label for masked images on y axis. We find, unsurprisingly, that adding TV regularization and upsampling make the mask more continuous and "human interpretable" and, more importantly, make it harder to find masks that can get high probability for the second best label, thus ensuring higher soundness.

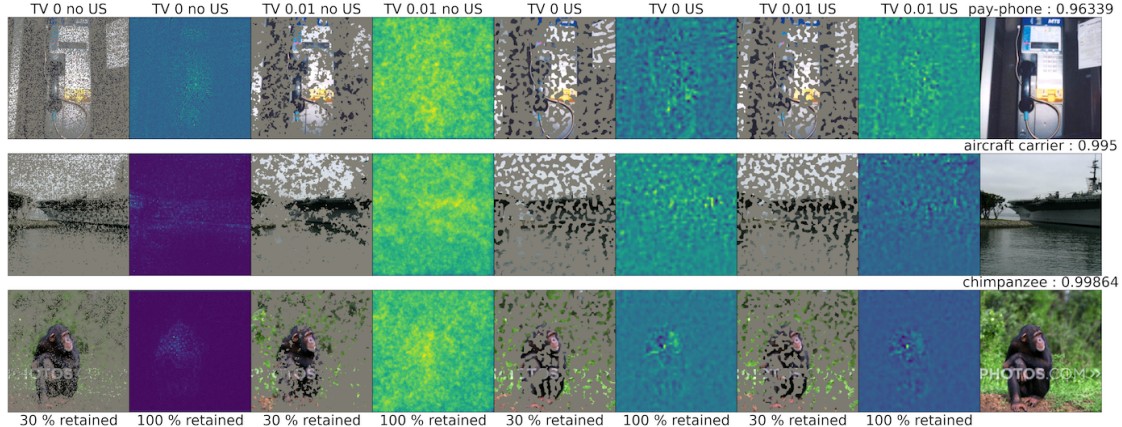

Figure 15: Results of randomizing the last layer of a ResNet-18 model on ImageNet data for the procedure described in Section 4. US indicates a $(56, 56)$ map was learned and upsampled to $(224, 224)$. We find the maps of this randomized network are less visually coherent than the analogous maps of a pre-trained model.

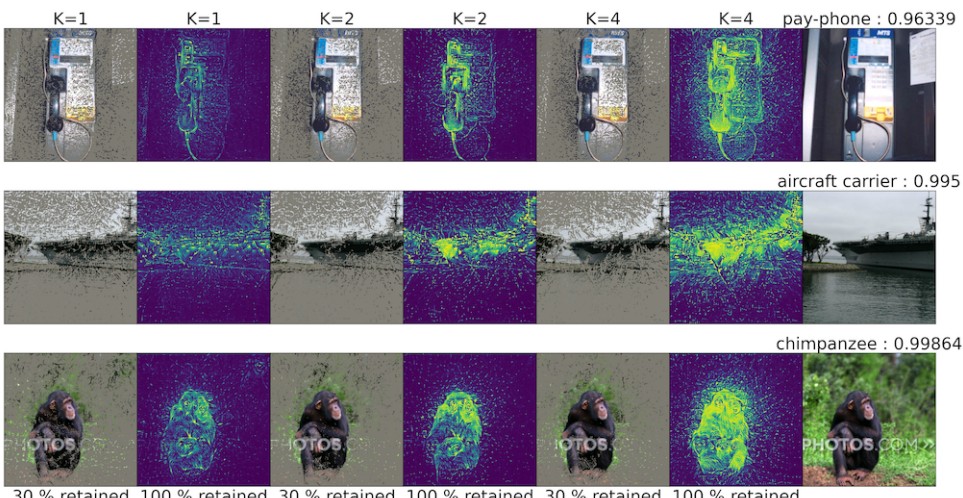

Figure 16: **Effect of ensembling** Partial statistical assignments validating the correct label of ImageNet and ResNet-18 images as we vary K, the number of maps. Details in Appendix D.3.

# E   Additional Background information

## E.1   Additional related work

The pixel replacement strategy we used is closely related to hot deck imputation [32], where features may be replaced either by using the mean feature value (analogous to replacing with grey) or sampling from the marginal feature distribution (analogous to replacing with image pixels sampled from other training images). Some prior work [7] has found that mean imputation does not significantly affect model output on the beer aroma review dataset. On Imagenette, by contrast, we found that replacement strategy can matter.

**Additional saliency evaluation tests.**   Adebayo et al. [1] proposed "sanity checks" for saliency maps. They note that if a model's weights are randomized, it has not learned anything, and therefore the saliency map should not look coherent. They also randomize the labels of the dataset and argue that the saliency maps for a model trained on this scrambled data should be different than the saliency maps for the model trained on the original data. We study the effect of model layer randomization on our method in Appendix section D.4 and find that randomizing the model weights does cause our

saliency maps to look incoherent. Tomsett et al. [43] also discover sanity checks for saliency metrics, finding that saliency evaluation methods can yield inconsistent results. They evaluate saliency maps on reliability, i.e. how consistent the saliency maps are. To measure a method's reliability, because access to ground truth saliency maps are not available, they use three proxies 1) inter-rater reliability, i.e. how whether a saliency evaluation metric is able to consistently rank some saliency methods above others, 2) inter-method reliability, which indicates whether a saliency evaluation metric agrees across different saliency methods, and 3)internal consistency reliability, which measure whether different saliency methods are measuring the same underlying concept.

**Saliency axioms.** Sundararajan et al. [41] identify two fundamental axioms, Sensitivity (versions a and b), and Implementation Invariance that attribution methods should satisfy. "An attribution method satisfies Sensitivity(a) if for every input and baseline that differ in one feature but have different predictions then the differing feature should be given a non-zero attribution." The definition of Sensitivity b is "If the function implemented by the deep network does not depend (mathematically) on some variable, then the attribution to that variable is always zero.". Implementation invariance simply states that if two networks are equivalent, the saliency maps for those two networks should be the same. These axioms are not captured by completeness and soundness and are good examples of why completeness and soundness cannot be used alone in evaluating saliency maps. Other prior work Carter et al. [7] argues that saliency explanations should be minimal Carter et al. [7] and find sufficient input subsets which are "minimal subsets of features whose observed values alone suffice for the same decision to be reached, even if all other input feature values are missing."

## E.2   Saliency Methods

We give a partial list of extant saliency methods here. We broadly categorize explanations into three categories: Back-propagation based explanations, axiomatic methods, and masking methods. **Backpropagation based explanations** shape credit as it is propagated backwards through the neural network according to certain rules. These approaches include **Layerwise Relevance Propagation** [5] which satisfies completeness, **Rect-Grad** which thresholds internal neuron activations [21], and **DeepLIFT** which satisfies the summation to delta rule.

**Axiomatic methods.** Axiomatic methods decompose the ouput (typically the logit) according to certain axioms like fairness in Shapley based methods **SHAP** [26] and **conceptSHAP** [45]. We also include gradient based approaches like **Gradient** $\frac{\partial S}{\partial x}$ [3] which calculates the partial derivative of the logit with respect to the input. **Gradient** $\odot$ **Input** [38] $\frac{\partial S}{\partial x} \cdot x$, which elementwise multiplies the gradient explanation by the input, and **Grad-CAM** [36] which takes the gradient of the logit with respect to the feature map of the last convolutional unit of a DNN. **Smooth-Grad** [39], which averages the **Gradient** $\odot$ **Input** explanation over several noisy copies of the input $x + \eta$, where $\eta$ is some Gaussian. The previous methods are intrinsic in the sense that they aim to explain the model decision. The last category of saliency maps, namely masking methods, also aim to explain the model decision, but they frequently aim to do so in a way that is interpretable by a human. Contrastive methods, such as contrastive layerwise propagation Gu et al. [13], also modify LRP by constructing class specific saliency maps, with the goal of object localization, i.e. in an image of an elephant and zebra, the saliency map for elephant should have high overlap with the elephant, and similarly for the corresponding map for zebra.

**Masking Methods.** Masking Methods are often evaluated using a pointing game or WSOL metric, which measures overlap with human labeled bounding boxes or explanations. These masking methods include techniques based on averaging over randomly sampled masks [30], optimizing over meaningful mask perturbations [11], and real time image saliency using a masking network [9]. Pixels that have been removed from the image by the mask may be replaced by graying out, by Gaussian blurring as in Fong and Vedaldi [11], or with infillers such as CA-GAN [46] used in Chang et al. [8], or DFNet [14]. De Cao et al. [10] find masks using differentiable masking. Taghanaki et al. [42] introduce a method that results in more accurate localization of discriminatory regions via mutual information.

**Pruning and information theory** Khakzar et al. [19] improve attribution via pruning. Schulz et al. [35] improve attribution by adding noise to intermediate feature maps.

**Saliency and Boolean Logic.** Previous work has also drawn connections between saliency and notions in logic. Ignatiev et al. [18] relates saliency explanations and adversarial examples by a generalized form of hitting set duality. Ignatiev et al. [17] develops a constraint-agnostic solution for computing explanations for any ML model. Macdonald et al. [28] develop a rate distortion explanation for saliency maps and prove a hardness result. Mu and Andreas [29] find a procedure for explaining neurons by identifying compositional logical concepts. Zhou et al. [49] describe network dissection, which provides labels for the neurons of the hidden representations. We are unaware of frameworks like Section 3.

**Arguments about saliency.** For discussion including pro/cons of various methods some starting points are Seo et al. [37] Fryer et al. [12] Gu et al. [13] Sundararajan and Najmi [40].

**Phang et al. [31]** We describe separately the masking procedure used by Phang et al. [31]. They begin by taking a pretrained model on ImageNet. The masker has access to the internal representations of the pre-trained model, and tries to maximize masked in accuracy and masked out entropy. They do not provide the ground truth label to the masker.

### E.3 Saliency Evaluation Methods

Saliency evaluation methods attempt to evaluate the quality of a saliency map. Many interpret the heatmap values as a priority order of saliency. Extrinsic evaluation metrics include the **WSOL** metric, which aim to measure overlap of the saliency map with a human annotated bounding box and the **Pointing Game** metric proposed by Zhang et al. [47] in which a pixel count as a hit if it lies within a bounding box and a miss otherwise, and the metric is $\frac{\text{\# Hits}}{\text{\# Hits} + \text{\#Misses}}$. Other more intrinsic methods include early saliency evaluation techniques like **MorF** and **LerF** Samek et al. [33], which involve removing pixels either in the order of highest importance or lowest importance and observing the area of the resulting curve. **Insertion and Deletion Games** of Petsiuk et al. [30] uses this too. The deletion game measures the drop in class probability as important pixels are removed, while the insertion game measures the rise in class probability as important pixels are added. (Our AUC discussion in Section 3 relates to this.) **Remove and Retrain (ROAR)** is a saliency evaluation method proposed by Hooker et al. [15]. Input features are ranked and then removed according to a saliency map. A new model is trained on the modified training set, and a larger degradation in accuracy on the modified test set compared to the original model on the original test set is regarded as a better saliency method. (NB: retraining makes this a non-intrinsic method.) Previous work has also introduced datasets specifically designed to test saliency methods. **BAM** Yang and Kim [44] creates saliency maps by pasting object pixels from MSCOCO Lin et al. [25] into scene images from MiniPlaces Zhou et al. [48]. The **Saliency Metric** proposed by Dabkowski and Gal [9] thresholds saliency values above some $\alpha$ chosen on a holdout set, finds the smallest bounding box containing these pixels, upsamples and measures the ratio of bounding box area to model accuracy on the cropped image, $s(a, p) = \log(\max(a, 0.05)) - \log(p)$ where a is the area of the bounding box and p is the class probability of the upsampled image.

### E.4 Saliency computations and underlying meanings of saliency

For simplicity this discussion assumes the datapoints are images and the classifier is a deep net. The heatmap in the saliency method is trying to highlight the contribution of individual pixels to the final answer. This is analogous to how a human may highlight relevant portions of the image with a plan. (Classic saliency methods in vision are inspired by studies of human cognition.) Saliency methods operationalize this intuitive definition in different ways, and we try to roughly categorise these as follows.

**Variational interpretation.** These interpret saliency in terms of effect on final output due to change in a single pixel –captured either via partial derivative of output with respect to pixel value (i.e., effect of infinitesimal change), or via change of output when this pixel is set to 0 or to a random (or "gray") value. Examples include **Gradient**, **Gradient** $\odot$ **Input** Shrikumar et al. [38], **Occlusion**

**Credit attribution guided by gradient.** These use the gradient to guide the assignment of saliency values. The gradient is interpreted as propagating values from the output to the input layer, and the values are partitioned/recombined at internal nodes of the net following some conservation principles. A key goal is to ensure *completeness*, which means that the sum of the attributions equal the logit value. Examples include **LRP**, **DeepLIFT** Shrikumar et al. [38], **Rect-Grad** Let $a_i^l$ be the activation

of some node in layer $l$, and $R_i^{l+1}$ be the backpropagated gradient up to $a_i^l$. Rect-grad replaces the vanilla chain rule, $R^l = \mathbf{1}[a_i > 0]$ with the rule that $R_i^l = \mathbf{1}[R_i^{l+1}a_i > \tau]$ for some threshold $\tau$. Hence, during a backward pass preference is given to nodes with large margin.

**Ensembling on top of above two ideas.** Ensembling methods combine saliency estimates over multiple inputs an an attempt to reduce noise in the final map. Examples include **Smooth-Grad**, Occlusion based methods, etc. We also include **Shapley Values** in this list.

The Shapley value aims to fairly distribute credit among a coalition of $N$ players. In the context of image saliency, each coordinate of the image input may be seen as a player, and the Shapley value computes $\sum_{S \subseteq N} \setminus \{i\} \frac{|S|!(n-|S|-1)!}{n!}(v(S \cup \{i\}) - v(S))$. It can be interpreted as the marginal contribution of player i, over all possible orderings of the coalition. In this sense, it can be seen as an ensembling method, as it averages over all possible random permutations.

**Analysis of saliency methods.** Previous work has analyzed ensembling methods like Smooth-grad, and found that it does not smooth the gradient Seo et al. [37]. They conclude that Smooth-Grad does not make the gradient of the score function smooth. Rather Smooth-grad is approximately the sum of a standard saliency map and higher order terms and the standard deviation of the Gaussian noise. It has also been found that Shapley values, despite having a uniqueness result, can differ in the way they depend on the model, data, etc Sundararajan and Najmi [40]. Fryer et al. [12] highlight several nuances that should be taken into account when considering Shapley values. They introduce Shapley values as averaging over submodels, and note that "the performance of a feature across all submodels may not be indicative of the particular performance of that feature in the set of optimal submodels.". They provide specific cases where satisfying the axioms of Shapley values works against the goal of feature selection.

# F   Clarifying benefit of TV regularization

This section gives more details of the discussion in Section 5 about how TV regularizers help ensure soundness even in a linear setting.

Let $\mathcal{S}$ be a dataset of labeled data $(\boldsymbol{x}, y)$ where the inputs are of unit norm and labels are binary, i.e., $\|\boldsymbol{x}\|_2 = 1$, $y \in \{\pm 1\}$. The model in question is a linear classifier $f(\boldsymbol{x}) := \text{sgn}(\langle \boldsymbol{w}, \boldsymbol{x} \rangle)$ parameterized by the weight vector $\boldsymbol{w} \in \mathbb{S}^{d-1}$, and it achieves the perfect accuracy on the set $\mathcal{S}$ with a margin $\gamma := \min_{(\boldsymbol{x},y) \in \mathcal{S}} y \langle \boldsymbol{w}, \boldsymbol{x} \rangle > 0$. We assume that the coordinates of $\boldsymbol{x}$ and $\boldsymbol{w}$ are uniformly bounded by $\frac{10}{\sqrt{d}}$, i.e., $\|\boldsymbol{x}\|_\infty \leq \frac{10}{\sqrt{d}}$, $\|\boldsymbol{w}\|_\infty \leq \frac{10}{\sqrt{d}}$ (10 can be changed to any other constant).

Let $\Gamma$ be the input modification process that sets all non-salient pixels to 0. We are interested in binary heatmaps, i.e., $\boldsymbol{m}$ assigns 1 to pixels in some salient set $S$, and 0 otherwise. According to Definition 3.3, $g(x, a, \boldsymbol{m}) = \mathbb{E}_{\tilde{x} \sim \Gamma(x, \boldsymbol{m})}[\mathbb{1}_{[f(\tilde{\boldsymbol{x}})=a]}]$. A simple calculation shows that this expectation is equal to $\mathbb{1}_{[a \sum_{i \in S} w_i x_i > 0]}$, and thus the goal is to find $S$ so that $a \sum_{i \in S} w_i x_i > 0$.

As we do not consider the full salient set informative, we are interested in salient sets with size constraint $|S| = L$ for some $1 \leq L \leq d$. There is a simple saliency method that achieves this goal: Given an input $\boldsymbol{x}$ and a label $a \in \{\pm 1\}$, sort the coordinates according to $a w_i x_i$ and take the highest $L$ coordinates as the salient set $S$.

It is easy to see that this method always produces $S$ with $a \sum_{i \in S} w_i x_i > 0$. Letting $a = y$ proves the completeness. However, this method does not satisfy soundness: a salient set $S$ with $a \sum_{i \in S} w_i x_i > 0$ can also be found for $a \neq y$!

Now we see how the TV constraint helps to ensure soundness (with good probability). A vector can be seen as a 1D image, and the TV of a salient set $S$ can be defined by $\text{TV}(S) := \sum_{i=1}^{d-1} |\mathbb{1}_{[i \in S]} - \mathbb{1}_{[i+1 \in S]}|$. For simplicity, we consider salient sets with TV at most 2. This means $S$ is just an interval. Given the size and TV constraints $|S| = L$, $\text{TV}(S) \leq 2$, it is easy to come out with the following saliency method: search over all the intervals of length $L$ and if an interval $S$ satisfies $a \sum_{i \in S} w_i x_i > 0$, return it as the salient set. Fortunately, this method does satisfy both completeness and soundness, as is justified by Theorem 5.1 in Section 5.

**Theorem 5.1.** *For $(\boldsymbol{x}, y) \in \mathbb{R}^d \times \{\pm 1\}$ with $\|\boldsymbol{x}\|_2 = 1$, after random shuffling of the coordinates, the following holds for any $L_1 = \Omega(\frac{1}{\gamma^2} \log \frac{1}{\delta})$, $L_2 = \Omega(\frac{1}{\gamma^2} \log \frac{d}{\delta})$:*

1. *(Completeness) With probability $1 - \delta$, there is an interval $\boldsymbol{m}$ of length $L_1$ s.t. $g(x, y, \boldsymbol{m}) = 1$;*

2. *(Soundness) With probability $1 - \delta$, $g(x, -y, \boldsymbol{m}) = 0$ holds for all intervals $\boldsymbol{m}$ of length $L_2$.*

*Proof.* Let $\boldsymbol{m}$ be any fixed interval of length $L$, associated with salient set $S$. The distribution of $\sum_{i \in S} w_i x_i$ is identical to the distribution of the sum of $L$ samples drawn from $\{w_1 x_1, \ldots, w_d x_d\}$ without replacement. Note that $dy w_1 x_1, \ldots, dy w_d x_d$ are $d$ numbers with mean $\gamma$, and their absolute values are bounded by $10^2 = O(1)$. By Chernoff bound,

$$\Pr\left[\frac{1}{L}\sum_{i \in S} dy w_i x_i \leq \gamma - \epsilon\right] \leq e^{-\Omega(\epsilon^2 L)}.$$

Set $\epsilon = \gamma$ ensures that $y \sum_{i \in S} w_i x_i > 0$ with probability $1 - e^{-\Omega(\gamma^2 L)}$. We can fix any interval $\boldsymbol{m}$ with $L = L_1$ to prove Item 1.

Taking union bounds over all intervals of length $L$, we can see that the probability of existing an interval of length $L$ that certifies $-y$ should be no greater than $\sum_{|S|=L} e^{-\Omega(\epsilon^2 L)} \leq d^2 e^{-\Omega(\gamma^2 L)}$. Setting $L = L_2$ proves Item 2. □

This shows that such salient sets make sense to humans: if the model predicts $y$, then we can find an interval of length $\tilde{\Omega}(1/\gamma^2)$ so that computing the inner product only in that interval leads to the same prediction; otherwise if the model does not predict $y$, such interval cannot be found. Thus it is sufficient to convince humans that the model predicts $y$ by only revealing the existence of such interval and the coordinate values in it.