# OpenReview forum: "New Definitions and Evaluations for Saliency Methods: Staying Intrinsic, Complete and Sound"
_NeurIPS.cc/2022/Conference — NeurIPS 2022 Accept_

### Official Review · Reviewer_Ze6X · 2022-07-11

**Rating:** 8
**Confidence:** 3
**Soundness:** 4 excellent
**Presentation:** 4 excellent
**Contribution:** 3 good

**Summary:**

The paper introduces and formalizes new evaluation metrics to ensure goodness of saliency methods, based on the logical concepts of completeness and soundness. The first ensures that the network's output is unchanged when using the masked (with the saliency map) input in place of the full image, which is what most of current evaluations methods for saliency methods require. The latter requires verifying that the same saliency method cannot be used to produce masked input that make the net output a different label, and therefore ensures that the evaluation of saliency maps appropriately track the model's probability of assigning labels.

**Questions:**

The paper is very complete and accurate, and I don't have main questions or suggestions. Minor ones are:
- In "Prior approaches" section, a number of extrinsic evaluation metrics are listed. Are there some intrinsic methods previously proposed?
- At line 207, according to Fig. 2, M2 AUC score should be 0.29 and not 0.37.
- In fig.4 caption, I think there is a mistake, since TV penalty factor Lambda_TV in line 5 should be 0.001 and not 0.01.



**Limitations:**

I think sharing code and data related to the paper would be beneficial for the scientific community.

**Strengths And Weaknesses:**

The paper's contributions are clear and significant, and explained in a straightforward and accurate manner. Examples are significant and useful. The originality of the contribution lies in connecting the context of saliency methods to logical proof systems and to formalize an evaluation approach which overcomes limitations of current methods and helps making them more rigorous and theoretically grounded. A simple saliency method based on optimization is proposed, which thanks of a change in the pixel replacement strategy allows to satisfy soundness at a small price in completeness. This is proven to work as expected when validated on various datasets and compared to other saliency methods. Furthermore, thanks to their formal frameworks of definitions, authors provide an intrinsic justification about why methods used heuristically to improve the aspect of masks (TV regularization and upsampling) actually work, in that they improve the soundness.

---

### Official Review · Reviewer_kmbL · 2022-07-11

**Rating:** 6
**Confidence:** 3
**Soundness:** 3 good
**Presentation:** 3 good
**Contribution:** 3 good

**Summary:**

This paper presents a method for attributing saliency (in the sense of determining which pixels contribute to a classification outcome) to an image. It does so in a novel fashion that explores the tradeoff between the notions of completeness and soundness, pointing out that prior work in this domain does not address the latter.

**Questions:**

Figures 1 and 2 present a good case for the issues addressed by the paper. There are many quantitative results included as well. I would have liked to see some additional results on the qualitative side if possible but also understand that there are constraints to the size of the manuscript.

Can the authors comment on the value of a measure of saliency being intrinsic rather than guided by e.g. an external neural network? I see merits for this, but would appreciate a succinct statement of why things should inherently be intrinsic.

Can the authors comment on what is the right tradeoff between completeness and soundness in interpretation since there is an ability to create an arc of success on both fronts. E.g. how should one adjust things to exert the optimal tradeoff for human interpretability?

**Limitations:**

There are not strong societal impacts of this work insofar as I can see and to the extent that these do exist the authors have made a good case.

**Strengths And Weaknesses:**

The paper itself is reasonably well written albeit with some typos (e.g. completeness is spelled wrong in some places). I find that this addresses an original angle of this type of assessment of how the neural network makes its determination and in a principled way that gives it an advantage over some of its predecessors. I think this is a significant result and generally view the conclusions drawn by this paper as positive.

---

### Official Review · Reviewer_9U1L · 2022-07-11

**Rating:** 8
**Confidence:** 2
**Soundness:** 4 excellent
**Presentation:** 3 good
**Contribution:** 4 excellent

**Summary:**

This paper presents an additional dimension, *soundness*, for evaluating saliency methods for explainable AI. The authors define this concept, then use it to provide both explanations for why existing heuristic methods work, and to suggest new saliency methods.


**Questions:**

- Lines 44-46 clearly describe the "mask shape" problem. The method introduced in Section 4 mitigates this problem by using pixels from a random image rather than grey pixels or other methods. Does this solve the mask shape problem, or just make it more difficult to solve? The new method requires solving a difficult segmentation problem (that I'm not sure humans would be very good at) but one that is nevertheless solvable.
- Lines 166-168: shouldn't other values for label $a$ enter in here? For example, Figure 3 (right) is for the second best label. Shouldn't there be an index over labels other than $a$?
- Line 184: Should be $\beta$ not $\alpha$.
- Are completeness and soundness similar to (or perhaps, the same as) sensitivity and specificity? If so, mentioning this would help to improve the clarity.
- Figure 4 typo "completenese"

**Limitations:**

The authors have adequately addressed limitations.

**Strengths And Weaknesses:**

### Strengths

I find this to be a useful and convincing paper. The paper is well written, but the presentation of the concepts could be made more crisp in parts (see questions, below).

### Weaknessess

Nothing major I could see, but this is somewhat outside my area.

---

### Official Review · Reviewer_Ea2U · 2022-07-12

**Rating:** 7
**Confidence:** 4
**Soundness:** 3 good
**Presentation:** 4 excellent
**Contribution:** 3 good

**Summary:**

New Definitions and Evaluations for Saliency Methods introduces intrinsic evaluation metrics for saliency methods --- completeness and soundness --- that do not require additional models or human evaluation. These metrics are grounded in logical proof concepts and force the method to output a saliency map that only explains the class of interest. The paper proposes a mask-based saliency method that optimizes for soundness as well as completeness. Evaluations compare the proposed saliency method to other mask-based saliency methods on soundness and completeness, deletion and insertion game metrics, and saliency metric.

**Questions:**

See the weaknesses section for questions and suggestions that would strengthen the paper.

**Limitations:**

Please include a discussion of limitations. Some questions I had were:
* What is the tradeoff between intrinsic and extrinsic evaluations? Do they both have a place in evaluating saliency methods, or are intrinsic evaluations like yours always better?
* Is there a tradeoff between completeness and soundness? Should we optimize for both equally, or is there ever a case where we should prioritize one over another? Can looking at completeness and soundness separately tell us anything different than looking at them together?

**Strengths And Weaknesses:**

**Strengths**
* *Intrinsic evaluation methods for saliency methods* --- This paper proposes that saliency methods should be evaluated on completeness and soundness. These attributes are grounded in logical proof systems and defined for saliency methods mathematically and textually in the paper. These concepts provide a formal intrinsic framework to evaluate saliency methods without requiring human evaluations. Human evaluators often measure how well the saliency maps to their representations, which may not always align with the model's representations. Intrinsic evaluations are better suited to evaluating saliency methods on their ability to reflect the underlying model.
* *Defining saliency method requirements* --- The paper reframes saliency by introducing completeness and soundness as two necessary constraints for saliency methods. Previously we only required completeness; saliency justified a model's prediction. By requiring completeness and soundness, saliency justifies a model's prediction but can not justify any other possible prediction. These requirements improve the specificity of what a saliency method should output and make it more straightforward to interpret the results of a saliency method. It also ensures that the saliency map for each class is distinct, which can improve our ability to compare maps and draw meaningful insight between possible predictions.
* *Clarity* --- The paper is very well written and precise. Section 3 is straightforward to follow, despite presenting complex definitions.

**Weaknesses**
* *Novelty of replacement strategy* --- The key novelty of the saliency method is its pixel replacement strategy, where a new pixel value is sampled from another random image. This strategy is known as hot-deck imputation, where replacement values are sampled from the marginal feature distribution. Existing work has used hot-deck imputation as a masking strategy. It has also shown hot-deck imputation and mean imputation (i.e., grey pixel replacement) result in similar changes to model outputs. (see "What made you do this? Understanding black-box decisions with sufficient input subsets" by Carter et al.). Given the similarities to this work, I suggest discussing it in the related work and including it in the comparison to existing metrics and methods.
* *Missing related work* --- Related work on saliency evaluation methods should include model and data randomization tests from "Sanity Checks for Saliency Maps" by Adebayo et al.. Also, consider the saliency method axioms from "Axiomatic Attribution for Deep Networks" by Sundararajan et al.. "Sanity Checks for Saliency Metrics" by Tomsett et al. has a good evaluation of existing saliency evaluations.
* *Lack of reproducibility* --- The checklist indicates the paper does not include the compute details, code, or data. Please include computing details and other details needed for reproducibility. If possible, also release the code.
* *Limited limitations section* --- The paper does not discuss limitations. Understanding limitations is essential for readers who are looking to use this work. Please include a discussion on important considerations when using your method. Also, please incorporate the ethical considerations in Checklist 1 in the main text.

**Minor Issues**
* Line 19: missing space between emdash and words
* Line 37: "and if so one" --> "and, if so, one"
* Line 213: "Procedures for find masking explanations" --> "Procedures for finding masking explanations"
* Line 278: "from original test set" --> "from the original test set"

---

### Author Response · Authors · 2022-08-02
**Author Response (Main Points)**

We thank the reviewers for their time and thoughtful reviews. We address the main points, which are that we did not discuss limitations of our work, we need to discuss the tradeoff between completeness and soundness, and need to cite additional related work.
Due to the 9 page limit, we have had to include some modifications in the Appendix, but will incorporate them in the main paper should the paper be accepted (and an additional content page allowed.)

Summary of main changes to paper:
1) We added Appendix A with discussion of limitations
2) We added Appendix F.1 with citations of missing work and discussion
3) We corrected all typos

**[Ea2U]:** Please discuss limitations of your work **Answer:** Thank you; we have added a discussion in our paper in **Appendix A** (due to space limitations) to the following effect: completeness and soundness do not involve or address human interpretability of explanations. This can be a strength when considering intrinsic evaluations, but is a weakness when considering extrinsic evaluations. Additionally, though completeness and soundness should be taken into account while designing a saliency method, they are not meant as a replacement for other evaluation methods, which can provide additional information on saliency map quality.


**[kmbL, Ea2U]:**  What is the tradeoff between completeness and soundness; should both be optimized or one over the other? **Answer:** Completeness and soundness measure two different concepts. Many current saliency methods are designed to satisfy completeness, i.e. they verify the masked input causes the net to output the same label as the full input. Soundness additionally requires verifying that the same saliency method cannot be used to produce masked inputs that make the net output a different label. (lines 55-57). We argue that both completeness and soundness are needed and both should be optimized when designing a saliency method. In our experiments ensuring soundness can somewhat hurt completeness Figure 3 (right).


**[Ea2U]:**  Missing related work on 1) Hot deck imputation 2) model and data randomization tests from "Sanity Checks for Saliency Maps" by Adebayo et al.. 3) saliency method axioms from "Axiomatic Attribution for Deep Networks" by Sundararajan et al..  4)"Sanity Checks for Saliency Metrics" by Tomsett et al. 5)"What made you do this? Understanding black-box decisions with sufficient input subsets" by Carter et al.). **Answer:** Thank you. We have included citations for 1) 2) 3) 4), and 5) with discussion in Additional Background Information in the **Appendix F.1** because of the page limit. We will move it to the main paper. We did examine the effect of the sanity checks on our method in Appendix E.4.   5) aims to find a minimal explanation. They favor the mean-imputation approach over sampling-based imputation (i.e. they favor replacing with grey.) While in their experiments on the aroma beer reviews training set, they find replacement strategy does not have a large effect on model output, we find that using different replacement strategies can change the result on Imagenette.

---

> ### Author Response · Authors · 2022-08-02
> **Author Response (Other Points)**
>
> **[kmbL, Ze6X, Ea2U]:**  Have prior intrinsic methods been proposed? Why should methods be intrinsic? What is the tradeoff between intrinsic and extrinsic evaluations? Succinct statement as to why intrinsic methods are important. **Answer:** As discussed in the paper, Intrinsic methods and evaluations have a long history. Many of the early saliency methods are derived from an axiomatic approach (Gradient*Input, LRP, Shapley etc.) and hence intrinsic. But they do not perform well on more recent evaluations. Some existing *evaluations* such as insertion game and saliency metric are intrinsic. Succinctly: Intrinsic evaluation methods aim to quantify how well a saliency method explains the model’s decision, without bias towards explanations that match human intuition. Hence if explaining model decisions is a priority, intrinsic evaluations should be used. Extrinsic evaluations are important when a human overseer needs to understand the saliency heatmap, a setting which we do not study.
>
> **[91UL]:**  Are completeness and soundness the same as sensitivity/specificity? **Answer:** If the question refers to classic specificity/sensitivity of tests (wrt false positive, false negative etc.,) then that is somewhat reminiscent but not quite the same. Sensitivity refers to the probability of a positive test conditioned on having positive ground truth, while specificity refers to the probability of a negative test conditioned on having negative ground truth. In our case, the ground truth model probability may lie anywhere in [0,1], and hence we do not binarize to having a positive or negative ground truth. Rather, f(x,a) is the fractional truth value of statement (x,a). Completeness and soundness together ensure that the evaluation of the masked inputs produced using various labels approximately tracks this fractional ground truth. Sensitivity/specificity are similar to completeness/soundness in the sense that higher soundness can sometimes imply lower completeness (as in Fig 3.) Completeness and soundness also apply in the multi-class setting.
>
> On the other hand, if the question refers to sensitivity in context of saliency methods, namely “class sensitivity,” then they are not for the following reasons. (We are not sure what specificity means in this context.)  Class sensitivity rewards explanations that are significantly different between the highest probability and lowest probability label. This is often measured by taking the Pearson’s correlation between 1) the map for the highest probability label and 2) the map for the lowest probability label. This is not the same as checking whether a map can certify a low probability label into being a high probability label. E.g. one could construct a masked map having low correlation with 1) that also causes the net to output a different label.
>
> **[91UL]:** Lines 44-46 clearly describe the "mask shape" problem. Does using pixels from a random image solve the mask shape problem, or just make it more difficult? The new method from Sec 4 requires solving a difficult segmentation problem (that I'm not sure humans would be very good at). **Answer:** When the replaced pixels are grey, as Figure 1 depicts, the mask can ‘draw’ a different class on the image. Replacing with random image pixels may make it difficult for the saliency method to draw another class. Note that the goal is to maximize the probability that the neural net outputs a certain label, and indeed we should not expect humans to be able to solve this problem. (Please note again our discussion elsewhere that the intrinsic explanations are not geared towards the human overseer.)
>
> **[91UL]:** Lines 166-168: shouldn't other values for label “a” enter in here? For example, Figure 3 (right) is for the second best label. Shouldn't there be an index over labels other than “a”? **Answer:** Yes in our Definitions 3.1 and 3.2 we take “a” can to be any label and follow that convention in lines 166-168. Hence “a” does not have to be the second best label, and instead can be any label. In Figure 3 we showed the second best label as an example but any other label could be used.
>
> **[Ea2U, Ze6X, 91UL, kmbL]** All reviewers want code released
> **Answer:** We will gladly release code before camera-ready.
>
> ### Typos
> **[Ea2U]** Line 19: missing space between emdash and words
> **[Ea2U]** Line 37: "and if so one" --> "and, if so, one"
> **[Ea2U]** Line 213: "Procedures for find masking explanations" --> "Procedures for finding masking explanations"
> **[Ea2U]** Line 278: "from original test set" --> "from the original test set"
> **[Ze6X]** At line 207, according to Fig. 2, M2 AUC score should be 0.29 and not 0.37.
> **[Ze6X]** In fig.4 caption, I think there is a mistake, since TV penalty factor Lambda_TV in line 5 should be 0.001 and not 0.01.
> **[91UL]** Line 184: Should be  \beta not  \alpha.
> **[91UL]** Figure 4 typo "completenese"
>
>
>
> Answer: We have corrected all typos.

---

> > ### Comment · Reviewer_Ea2U · 2022-08-07
> > **Changes Look Good To Me**
> >
> > I want to thank the authors for responding to and incorporating suggestions from my fellow reviewers and me. I continue to recommend accepting this paper.

---

> > ### Comment · Reviewer_Ze6X · 2022-08-08
> > **Changes good**
> >
> > I think authors addressed reviewers comments in a satisfying way. I confirm my rating.

---

### Meta-Review · Area_Chair_qRPc · 2022-08-24

**Recommendation:** Accept
**Confidence:** Certain

**Metareview:**

The paper introduces and formalizes new evaluation metrics to ensure goodness of saliency methods,.

Reviewers consensus about the paper was positive. They found that the paper contributions are clear and significant and also appreciated the paper originality. I therefore recommend acceptance.



**Award:**

No

---

### Decision · Program_Chairs · 2022-09-14

Accept